# Building synthetic chromosomes from natural DNA

Alessandro L. V. Coradini [1] ✉, Christopher Ne Ville[1,2], Zachary A. Krieger[1,2], Joshua Roemer [1], Cara Hull [1], Shawn Yang [1], Daniel T. Lusk[1] & Ian M. Ehrenreich [1] ✉

De novo chromosome synthesis is costly and time-consuming, limiting its use in research and biotechnology. Building synthetic chromosomes from natural components is an unexplored alternative with many potential applications. In this paper, we report CReATiNG (Cloning, Reprogramming, and Assembling Tiled Natural Genomic DNA), a method for constructing synthetic chromosomes from natural components in yeast. CReATiNG entails cloning segments of natural chromosomes and then programmably assembling them into synthetic chromosomes that can replace the native chromosomes in cells. We use CReATiNG to synthetically recombine chromosomes between strains and species, to modify chromosome structure, and to delete many linked, non-adjacent regions totaling 39% of a chromosome. The multiplex deletion experiment reveals that CReATiNG also enables recovery from flaws in synthetic chromosome design via recombination between a synthetic chromosome and its native counterpart. CReATiNG facilitates the application of chromosome synthesis to diverse biological problems.

It is now possible to answer fundamental questions in biology by synthesizing chromosomes[1,2]. For example, a longstanding question has been what is the minimal set of genes required by a living cell[3–6]? To answer this question, researchers used design-build-test cycles to synthesize a *Mycoplasma mycoides* chromosome that contains only 473 genes and still produces a free-living bacterium that replicates on lab timescales[7]. Generating this minimal cell involved eliminating 428 (48%) of the genes that are naturally present in *M. mycoides*. Among the genes remaining in this minimal cell, 83% functioned in the expression and preservation of genetic information, the cell membrane, or cytosolic metabolism, while 17% had unknown functions. Production of this minimal cell demonstrates the potential for using chromosome synthesis to understand defining mechanisms underlying cellular life and its diversity.

To date, synthetic chromosomes have exclusively been generated de novo, through the progressive assembly of small synthetic DNA fragments into larger molecules via a combination of in vitro and in vivo techniques[7–22]. De novo synthesis is powerful because it allows the complete reprogramming of a chromosome's sequence and structure. For example, de novo chromosome synthesis was used to generate an *Escherichia coli* strain in which all 18,214 instances of three codons were synonymously reprogrammed, resulting in a strain that utilizes only 61 codons[18]. In another example, the Sc2.0 community is using de novo chromosome synthesis to generate a strain of the model budding yeast *Saccharomyces cerevisiae* in which all transposable elements have been eliminated and LoxP sites have been incorporated between genes, enabling the generation of random chromosome rearrangements by Cre recombinase[12].

The substantial amount of DNA fragment synthesis and assembly involved in de novo chromosome synthesis limits its use in biological research. Reductions in labor and reagent costs are needed to enable biologists to employ chromosome synthesis more widely. Building synthetic chromosomes from cloned segments of natural DNA could be a relatively cheap and fast alternative to de novo chromosome synthesis. Such a method would enable the use of chromosome synthesis in research that does not require complete chromosome reprogramming. For example, projects that could be

[1]Molecular and Computational Biology Section, Department of Biological Sciences, University of Southern California, Los Angeles, CA 90089, USA. [2]These authors contributed equally: Christopher Ne Ville, Zachary A. Krieger. ✉e-mail: coradini@usc.edu; ian.ehrenreich@usc.edu

enabled include mapping the genetic basis of trait differences between individuals and species, probing the structural requirements of chromosomes, and streamlining chromosomes through the systematic elimination of non-essential genetic elements.

In this paper, we introduce CReATiNG (Cloning, Reprogramming, and Assembling Tiled Natural Genomic DNA), a method for building synthetic chromosomes from natural components in *S. cerevisiae*. The first step of CReATiNG is the cloning of natural chromosome segments such that unique adapter sequences are appended to their termini, specifying how these molecules will recombine with each other later when they are assembled. The second step of CReATiNG is co-transforming cloned segments into cells and assembling them by homologous recombination in vivo. Synthetic chromosomes generated with CReATiNG can replace the native chromosomes in cells, making it possible to directly test their phenotypic effects. Here, we describe the steps involved in CReATiNG and demonstrate several of CReATiNG's use cases.

## Results

### A system for cloning and reprogramming natural DNA for assembly

CReATiNG involves cloning segments of natural chromosomes in yeast donor cells and then programmably assembling these segments into

synthetic chromosomes in different recipient cells. To clone a target segment, we co-transform three reagents into donor cells that constitutively express Cas9 (Fig. 1a, b; Supplementary Fig. 1; Supplementary Table 1): (1) in vitro transcribed (IVT) guide RNAs (gRNAs) that direct Cas9 to cut a target segment on each side, excising it from a chromosome; (2) a linear Bacterial Artificial Chromosome/Yeast Artificial Chromosome (BAC/YAC) cloning vector (pASC1) flanked by homology to the ends of the segment, enabling integration of the segment into the vector in vivo by homologous recombination; and (3) a repair template comprised of a dominant drug marker (*KanMX*) flanked with homology arms that allow a cell to reconstitute its broken chromosome by homologous recombination, resulting in replacement of a segment with a marker.

In this study, Cas9 was expressed from a strong constitutive promoter (*TDH3*) off a plasmid (pML104)[23] carrying the *HIS3* marker (Supplementary Table 1). We transformed pML104 into a donor yeast strain prior to attempting any cloning reactions. We used IVT gRNAs instead of more conventional gRNA plasmids because the former allows PCR products to be used as templates, eliminating assembly and verification of gRNA plasmids[24]. IVT gRNAs also make it possible to use several distinct gRNAs for the same target in the same transformation with little added work, removing the need to screen individual gRNAs for efficacy. We employed three gRNAs for each side of a target

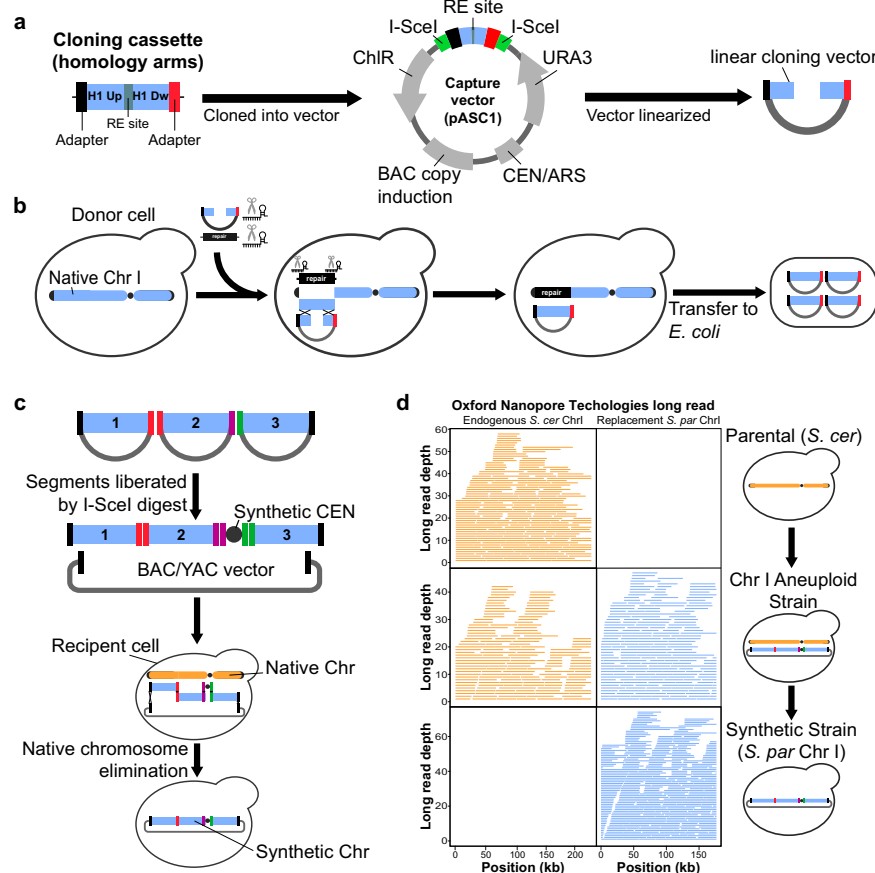

**Fig. 1 | Synthesizing chromosomes from natural components in yeast using CReATiNG. a** The Bacterial Artificial Chromosome/Yeast Artificial Chromosome (BAC/YAC) vector used for cloning natural chromosome segments in vivo. Homology may be flanked by sequence adapters that program how a segment will assemble with others in later steps. **b** A segment is cloned by co-transforming a linearized cloning vector, gRNAs targeting both sides of the segment, and a selectable repair template into a donor cell constitutively expressing Cas9. Cloned segments are then extracted from yeast and transferred to the *E. coli*. **c** Cloned segments are excised from the vector through restriction digestion with I-SceI.

These molecules are then purified and co-transformed into a recipient yeast cell with a centromere cassette and a centromere-free version of the BAC/YAC cloning vector. These molecules are assembled into a synthetic chromosome by homologous recombination in vivo, while the native chromosome is eliminated by centromere destabilization and counterselection. **d** ONT sequencing confirmed the correct assembly of *S. paradoxus* ChrI (blue) and replacement of the native *S. cerevisiae* ChrI (orange) in BY. The plot shows reads mapped to each chromosome using a reference genome including both BY and *S. paradoxus* ChrI.

segment (six total per segment), choosing gRNAs with the highest predicted on-target scores within 1 kb of the homology sites used for cloning. If genes were present within the 1 kb regions, preference was given to gRNAs that had lower on-target scores but were outside known coding or regulatory regions. The maximum distance between gRNAs targeting the same side of a segment was ~200 bp.

pASC1 contains standard BAC/YAC components, such as most of the pCC1BAC copy control vector for *E. coli* and a portion of the *URA3*-marked yeast centromeric plasmid pRS316. In addition, pASC1 was engineered to be amenable to high-throughput integration of cassettes for cloning specific targets by directional restriction digestion and ligation. A cloning cassette is typically ~500 bp and contains two 120 bp homology arms that are specific to the termini of a segment, separated by AvrII and XhoI sites, and flanked by 100 bp DNA sequences (adapters) and I-SceI sites, which are motifs absent from the *S. cerevisiae* genome. Prior to cloning a given target segment, we synthesize its cloning cassette de novo and integrate the cassette into pASC1. After integration of a cloning cassette, the AvrII and XhoI sites between the homology arms facilitate linearization of pASC1 by restriction digestion, resulting in a linear molecule with incompatible sticky ends and a low chance of recircularization. The adapters are used to program how different segments will assemble later, as different segments with the same adapters will recombine when co-transformed into recipient yeast cells. In our experience, adapters are necessary for high-efficiency assembly of *Saccharomyces* chromosomal segments inside *Saccharomyces* cells. We generate adapters using a random sequence generator[25], restricting GC content to ≥40%. We only used random adapters if they lacked mononucleotide repeats of ≥7 bases and did not show detectable identity to the *S. cerevisiae* and *S. paradoxus* reference genomes. I-SceI sites enable a cloned segment to be liberated from the vector by restriction digestion.

As we show later in the Results, after transformation with the cloning reagents, donor cells containing successful cloning events can be isolated by selecting for the markers in the cloning vector (*URA3*) and the repair template (*KanMX*). Successful cloning can be verified by PCRs at the junctions between the vector and a cloned segment. Cloned segments can be extracted from yeast donor cells and transformed into EPI300 *E. coli* cells, which allow for induction of high pASC1 copy number. After induction, pASC1 containing a segment is then extracted from *E. coli* and the segment is separated from the vector by digestion with I-SceI. Multiple distinct segments that share the same terminal adapters can then be co-transformed into recipient yeast cells along with an appropriate vector, enabling selection for cells containing chromosome assemblies.

## Initial assembly of a chromosome using CReATiNG

To prototype CReATiNG, we used Chromosome I (ChrI), a 230 kb chromosome containing 117 known or predicted protein-coding genes[26]. ChrI is the smallest chromosome in the *Saccharomyces* genus and shows synteny between species[27]. We chose *S. paradoxus* (Supplementary Table 2) as the initial donor for prototyping CReATiNG because it shows ~12% nucleotide divergence from *S. cerevisiae*[28], making it possible to easily distinguish an assembled chromosome from the native chromosome in a recipient *S. cerevisiae* cell. In silico, we divided *S. paradoxus* ChrI into three non-overlapping segments between 51 and 64 kb, which contained the entire chromosome except the centromere, subtelomeres, and telomeres (Supplementary Fig. 2A). The segments were each designed to contain roughly one-third of ChrI and to avoid disruption of annotated functional elements. We cloned segments 2 and 3 in a manner that excluded the natural ChrI centromere, as we supply a synthetic centromere cassette containing *CEN6*, as well as drug resistance markers for both yeast (*KanMX*) and *E. coli* (*ampR*), during chromosome assembly. We excluded subtelomeres from all subsequent work, as they are completely dispensable and highly variable across *Saccharomyces* strains and

species[27]. Telomeres were also excluded because they are not amenable to cloning and we initially assemble chromosomes as circular, rather than linear, molecules.

To enable cloning of the *S. paradoxus* ChrI segments, we generated a Cas9-expressing version of the *S. paradoxus* CBS5829 strain by transforming it with *HIS3*-marked pML104. We then used this *S. paradoxus* strain that constitutively expresses Cas9 for three transformations, each designed to clone a different segment (Supplementary Table 3). For each target segment, we generated a distinct version of pASC1 containing appropriate homology arms and co-transformed it with six gRNAs and a repair template (Supplementary Table 4), as discussed in the previous section. For each transformation, we checked five random colonies by amplifying each junction between a cloned segment and the vector (Supplementary Figs. 2B–D; Supplementary Table 5). Based on amplification of both junctions and Sanger sequencing, 14 of 15 (93%) colonies showed successful cloning (Supplementary Table 5). After transfer to and amplification in *E. coli*, the three *S. paradoxus* ChrI segments were extracted, liberated from the vector by I-SceI digestion, checked for correct size on an agarose gel, and purified (Supplementary Fig. 3A).

Next, we assembled the segments in their natural order and orientations by co-transforming them, the centromere cassette with appropriate adapters, and a linear pASC2 (Supplementary Table 1), a version of pASC1 that lacks a centromere and contains *HIS3* as the yeast marker, into the BY4742 (BY) reference strain of *S. cerevisiae* (Fig. 1c). We programmed this assembly by placing distinct adapters between segments 1 and 2, segment 2 and the centromere cassette, and the centromere cassette and segment 3. After transformation, recipient cells in which the five molecules had assembled were isolated by selection on markers present in pASC2 and the centromere cassette. Of five colonies checked by PCR of junctions between assembled segments, four (80%) contained the complete assembly (Supplementary Table 7). We then performed whole genome Oxford Nanopore Technologies (ONT) sequencing of a single yeast clone and confirmed the presence of two copies of ChrI, one from BY and one from *S. paradoxus* (Fig. 1d).

To produce a euploid strain containing only *S. paradoxus* ChrI (IEY394), we conditionally destabilized and selected against BY ChrI[12,29] (Supplementary Fig. 3B). Chromosome elimination involves disrupting centromere function with an inducible *GAL1* promoter and counter-selecting a *URA3* marker on the chromosome using 5-FOA. These elements were engineered into the recipient strain prior to the chromosome assembly transformation. We verified complete elimination of BY ChrI by PCR (Supplementary Fig. 3C) and ONT sequencing (Fig. 1d). We also used the ONT data to confirm that the euploid strain IEY394 had the expected sequence and structure genome-wide, with the exception of a single point mutation in a non-functional portion of the centromere cassette (Supplementary Fig. 4). This mutation was likely introduced during the PCR amplification of the centromere cassette, indicating the cloning and assembly process used in CReATiNG does not exhibit detectable mutagenicity. Further, analysis of genome-wide sequencing coverage for IEY394 and BY indicated that *S. paradoxus* ChrI was present at a single copy per cell (Supplementary Fig. 5).

While the remaining work in this manuscript was conducted with circular chromosomes, some chromosome synthesis applications could require linear chromosomes. To confirm that it is possible to convert chromosomes synthesized by CReATiNG from circular to linear forms, we linearized the synthetic *S. paradoxus* ChrI. We used Cas9 to introduce double-strand breaks near each junction between the chromosome and the cloning vector. In addition to gRNAs, we co-transformed repair templates for both chromosome ends, each of which contained a synthetic telomere seed[30] and a distinct selectable marker. By selecting for the markers in both repair templates, we obtained cells containing a linear *S. paradoxus* ChrI (Supplementary Fig. 6).

## Recombination of chromosomes between strains and species

After confirming that CReATiNG can be used to build synthetic chromosomes that replace the native chromosomes in recipient cells, we explored potential applications. The first application was to synthetically recombine chromosomes between strains and species, which could aid efforts to study the genetic basis of heritable phenotypes. Relative to the crosses conventionally used to generate recombinants, the advantages of CReATiNG are that it does not require mating, meiosis, or natural synteny. Additionally, CReATiNG allows three or more parental chromosomes to recombine in a single assembly. The main constraint of CReATiNG for synthetically recombining chromosomes is that at present it cannot be applied genome-wide.

To use CReATiNG to recombine chromosomes synthetically, we next cloned the three ChrI segments from BY and another *S. cerevisiae* strain, the vineyard isolate RM11-1a (RM). During cloning, we appended the same adapters that were used for *S. paradoxus* segments, making it possible to generate all-possible syntenic combinations of the BY, RM, and *S. paradoxus* segments (Fig. 2a). As we had already generated a strain containing an entirely *S. paradoxus* ChrI, we individually assembled the 26 remaining possible chromosomes (Supplementary

Figs. 7 and 8). These assemblies had efficiencies between 20 and 100% based on PCR examination of five colonies per transformation (Fig. 2b; Supplementary Table 6). After elimination of native ChrI, we further verified assemblies by ONT sequencing (Supplementary Fig. 9 and Supplementary Table 7).

Next, we measured the growth rates of these strains with recombinant chromosomes (IEY394 to 420) in two conditions–rich liquid medium at 30 and 35 °C (Fig. 2c, d; Supplementary Table 9). *S. paradoxus* is known to be more sensitive to high temperatures than *S. cerevisiae*[31]. This phenotypic difference was also present in our donor strains (Supplementary Fig. 10). Among the 27 strains with recombinant chromosomes, growth rates varied substantially in both conditions (one-way ANOVAs, $p$ values $\leq 6.6 \times 10^{-11}$). In addition, the strain carrying a fully *S. paradoxus* ChrI exhibited the slowest growth at both temperatures, but the difference between it and other strains was more severe at higher temperature. Our results corroborate a recent finding that at least one locus on ChrI contributes to variation in thermotolerance between the two species[32].

The 27 strains with recombinant chromosomes also provide an opportunity to measure the individual and combined phenotypic

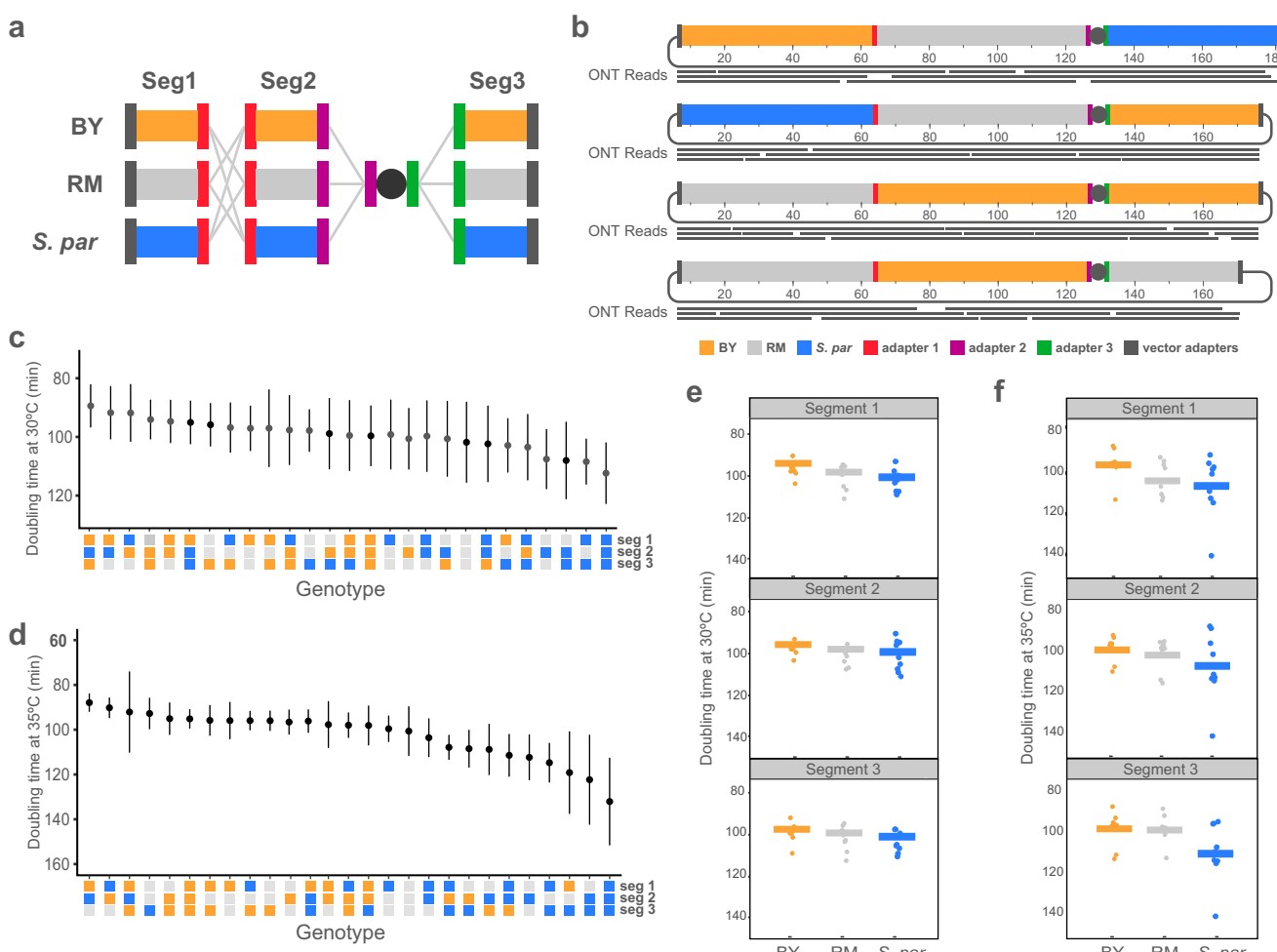

**Fig. 2 | Recombining chromosomes between strains and species using CReATiNG. a** Segments 1 through 3 were cloned from two additional strains, the BY (orange) and RM (gray) strains of *S. cerevisiae*. The 27 possible syntenic combinations of the *S. paradoxus* (blue), BY, and RM segments were then assembled in BY recipient cells and the native ChrI was eliminated. **b** Representative validations of assembled chromosomes by ONT sequencing. In silico designs of each chromosome are shown with a subset of mapped reads plotted below them. **c** and **d** Euploid strains carrying the synthetic chromosomes were phenotyped for doubling time in

rich medium containing glucose at 30 °C and 35 °C, respectively. For each strain, a total of 12 independent growth experiments were performed ($n = 12$). Each dot represents the mean of a genotype across the 12 replicates, with error bars representing one standard deviation around the mean. **e** The mean effect (horizontal line) of each segment across all genotypes at 30 °C is shown, with dots representing the mean of a genotype across 12 replicates. **f** The mean effect (horizontal line) of each segment across all genotypes at 35 °C is shown, with dots representing the mean of a genotype across 12 replicates.

effects of the three ChrI segments. We found that two segments had significant effects on growth at 30 °C (one-way ANOVAs, segment 1 $p$ value = 0.008 and segment 2 $p$ value = 0.019; Fig. 2e). However, at 35 °C, all three segments had highly significant effects (one-way ANOVAs, $p$ values $\leq 2.8 \times 10^{-5}$; Fig. 2f). These results show that all three ChrI segments contribute to growth variation across temperatures, with segment 3 in particular showing a strong interaction with temperature. We also used the data to measure epistasis among the segments and detected significant pairwise and three-way genetic interactions ($F$ tests, $p$ values $\leq 3.6 \times 10^{-4}$; Supplementary Table 10). Thus, thermotolerance differences between the species involve both additive and epistatic loci.

Because we only divided ChrI into three segments, our data cannot resolve causal genetic differences and must be regarded as an initial stage of genetic mapping, similar to quantitative trait locus mapping. However, consideration of our results in the light of prior work from other groups[31,32] and comparative genomic data suggests CReATiNG can produce new insights into the genetics of trait differences between species. Recent studies used reciprocal hemizygosity analysis to identify >50 genes that cause thermotolerance differences between *S. cerevisiae* and *S. paradoxus*. Among these genes, only one–the aminophospholipid translocase *DRS2*–resides on ChrI and is located in segment 2. By contrast, we found that all examined ChrI segments showed thermotolerance effects and the largest effect was due to segment 3. Another noteworthy result is that among the three segments, segment 3 is the only one that shows gene content differences between the species. In segment 3, relative to *S. cerevisiae*, *S. paradoxus* lacks the adjacent genes *DFP2* and *PRM9*, which encode DUP240 family proteins with unclear functions. Further, in *S. paradoxus* segment 3, *PAU12*, a protein of unknown function, is present at the location of *PAU7*. Higher resolution mapping is required for determining whether these genes or others play roles in interspecies differences in thermotolerance. Yet even at our presently crude mapping resolution, these findings show that CReATiNG is a useful tool for studying the contribution of genetic factors, including genotype-by-environment interactions and epistasis, to trait differences between strains and species.

## Chromosome restructuring

CReATiNG can also be used to experimentally probe the structural rules underpinning chromosome organization, a topic relevant to genome function and evolution. Recent work suggests yeast can tolerate a diversity of chromosome structures, but most of these studies preserved the order of naturally linked genes[33–36]. CReATiNG makes it possible to restructure the contents of a chromosome in specific non-natural configurations that are programmed using adapters. CReATiNG can be used to synthesize chromosomes with one or more inversions, duplications, deletions, or modifications to gene order.

To demonstrate how CReATiNG can be used in chromosome restructuring, we re-cloned segments 1 through 3 from BY. During this round of cloning, we modified the adapters appended to each segment, making it possible to assemble the segments in all possible orders without inverting any segment. Using these re-cloned segments, we designed five non-natural ChrI structures with the same content but different orders (i.e., 1-3-2, 2-3-1, 2-1-3, 3-1-2, and 3-2-1) (Fig. 3a). We produced euploid strains with each non-natural ChrI structure by assembling segments with appropriate adapters and then eliminating the native ChrI. Each assembly was verified by junction PCRs or ONT sequencing (Supplementary Fig. 11; Supplementary Table 11).

While all five strains possessing restructured versions of ChrI were viable, they also showed substantial phenotypic variation. The 2-3-1 (IEY423) configuration exhibited a 7% growth improvement relative to the natural 1-2-3 (IEY402) configuration, which had been generated earlier in the work on synthetic recombinants (Fig. 3b; Supplementary

Table 12). By contrast, the 3-1-2 (IEY421) and 1-3-2 (IEY425) configurations respectively showed growth reductions of 18% and 68% relative to the natural 1-2-3 configuration. Thus, relocating segment 2 to the natural position of segment 3 significantly impedes growth. However, the degree of this impairment also depends on the locations of segments 1 and 3.

Because the strains with restructured chromosomes all possess the same gene content, the likely explanation for these substantial growth defects is that chromosome restructuring created new genetic neighborhoods[37] at segment boundaries, resulting in the misexpression of phenotypically important genes. Evaluating all restructured chromosomes, unique junctions occur in the 3-1-2 and 1-3-2 strains that are not present in any other ChrI configurations (Fig. 3c). Several non-essential genes that are known to cause growth defects when overexpressed are located at these unique junctions: a type V myosin motor (*MYO4*), a vesicle membrane SNARE protein (*SNC1*), and a nucleoporin component (*NUP60*)[26]. Both strains possess a junction that places *MYO4* near the *KanMX* marker in the centromere cassette. We found that *MYO4* is 1.5- and 1.2-times more expressed on 1-3-2 and 3-1-2 strains, respectively, relative to the 1-2-3 (IEY402) control strain (t-test $p$ values < 0.05; Fig. 3d; Supplementary Tables 13–15). This result may help explain the 18% growth reduction in the strain 3-1-2 but by itself cannot explain the greater growth reduction in the strain 1-3-2 (68%). Thus, we also investigated gene expression of *SNC1* and *NUP60*, which are adjacent exclusively in the 1-3-2 strain. *SNC1* and *NUP60* are respectively 2.3- and 1.5-times more expressed in the 1-3-2 strain than the 1-2-3 control strain (t-test $p$ values < 0.05). These findings show how programmably restructuring chromosomes with CReATiNG can be used to identify non-natural chromosome configurations with phenotypic effects and suggest that changes in gene neighborhoods that cause gene overexpression will be a major source of growth defects in such experiments.

## Multiplex gene deletion and chromosome streamlining

Another application of CReATiNG is highly multiplexed deletion, a task that remains challenging for conventional genome editing technologies[38]. Multiplexed deletion could enable the generation of streamlined chromosomes in which many non-essential genetic elements have been eliminated[39]. Such streamlining can facilitate the production of yeast strains with a substantially reduced gene complement, as well as the reorganization of functionally related genes into modules. CReATiNG simplifies multiplexed deletion: segments of a natural chromosome that should be retained can be cloned and assembled, cleanly deleting all intervening parts of a natural chromosome.

We designed a CReATiNG workflow to delete ten core chromosome regions from BY ChrI, summing to 30 kb and containing 18 nonadjacent genetic elements in aggregate (12 protein-coding genes, 2 tRNAs, 3 LTRs, and 1 LTR retrotransposon; Fig. 4a; Supplementary Tables 16–18). We chose these elements because they are all annotated as non-essential and no synthetic lethal interactions have been reported among them[40]. This design required cloning 11 segments, which ranged in size from 3.8 kb to 21 kb (Supplementary Fig. 12A; Supplementary Tables 19 and 20), and programming them with appropriate adapters for assembly. During cloning, we made sure to avoid disrupting any annotated functional elements. In BY, the subtelomeres, which we also exclude from synthetic chromosomes, comprise 61.7 kb and 24 protein-coding genes, 1 pseudogene, and 2 LTRs (Supplementary Table 21). Thus, in total, our multiplex deletion design reduced ChrI by 39.9% (91.7 kb) and eliminated 45 genetic elements.

After cloning, we co-transformed the 11 segments into BY along with a centromere cassette and a linearized pASC2 vector. *ADE1*, a gene encoding an adenine biosynthesis enzyme whose loss of function causes colonies to accumulate red pigment, was present in one of the

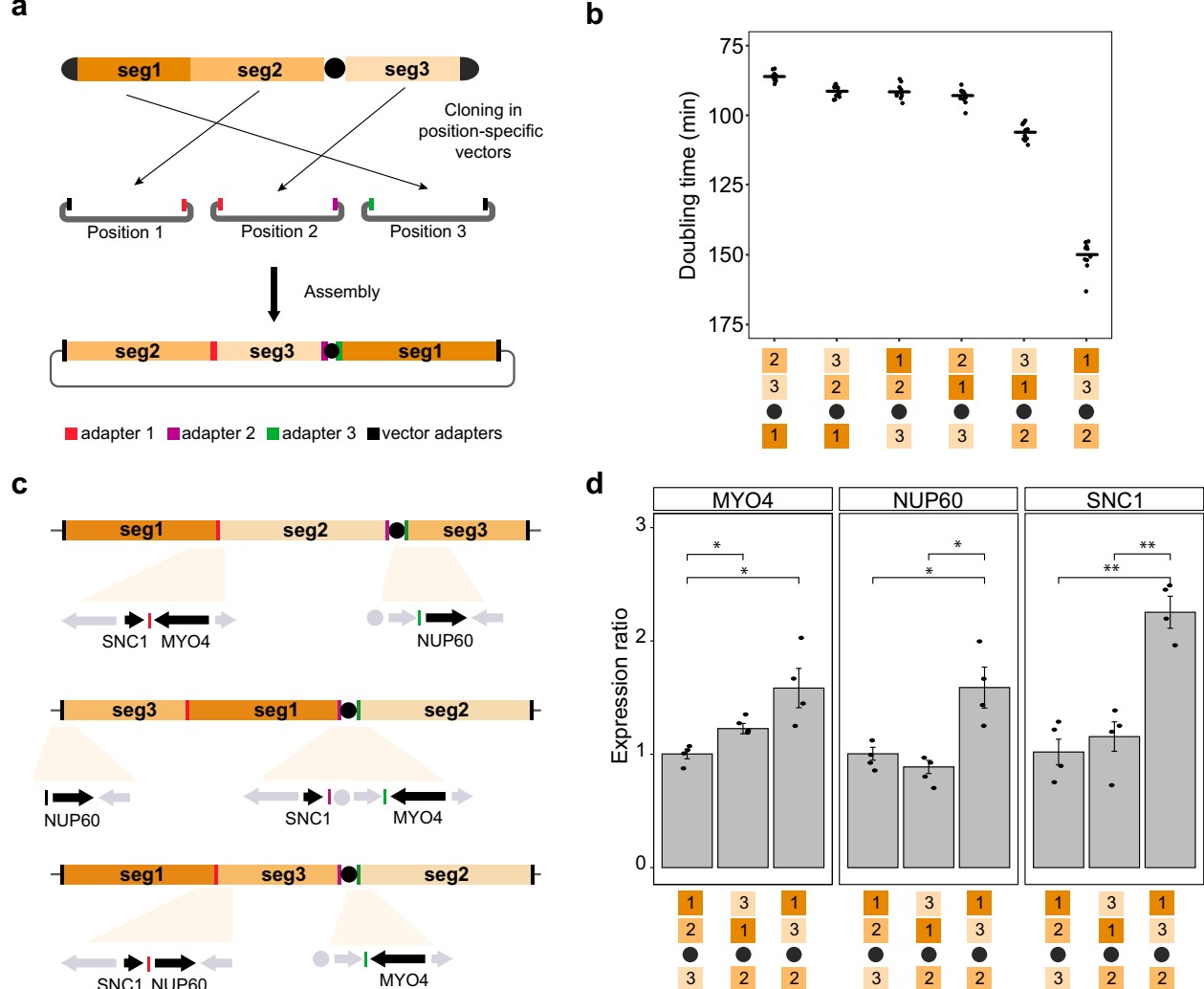

**Fig. 3 | Restructuring ChrI with CReATiNG. a** Chromosome I segments are cloned between a pair of position-specific adapters present into distinct versions of pASC1. Five possible restructured versions of BY ChrI were created by altering the adapters appended to each segment prior to their assembly. PCR of assembly junctions or ONT sequencing was used to confirm that the chromosome assemblies had the correct structure. **b** Growth analysis of the natural and five non-natural chromosome structures in a rich medium containing glucose at 30 °C. For each strain, a total of 12 independent growth experiments were carried out (*n* = 12). Each dot represents the value from an individual replicate of a given genotype, while the horizontal bar represents the mean across the 12 replicates. **c** The restructured ChrI containing the structures 1-3-2 and 3-1-2 present specific non-natural junctions involving the genes *SNC1*, *MYO4*, and *NUP60*. **d** Expression ratios of the genes *SNC1*, *MYO4*, and *NUP60* in the control strain IEY402 (1-2-3) and restructured strains IEY421 (1-3-2) and IEY425 (3-1-2) were calculated using the Pfaffl method[50] and *ACT1* as a control housekeeping gene. The values are the means and standard deviations of four independent biological replicates. Two-tailed t-tests with *, **, and *** corresponding to $p < 0.05$, $p < 0.01$, and $p < 0.001$, respectively. Exact *p* values are provided in Supplementary Table 15.

regions targeted for deletion and provided an extra phenotypic selection for native ChrI elimination. After native ChrI elimination, we picked ten red colonies and checked each at all assembly junctions by PCR (Supplementary Figs. 12B-D and 13). While none of the red colonies possessed all deletions, a single red colony had nine of the ten deletions, a finding confirmed by ONT sequencing (Fig. 4b). This strain with the multiplex deletion exhibited significantly slower growth than a strain with a synthetic BY ChrI lacking the deletions (212 and 94 min doubling times, respectively; t-test, *p* value = $6.1 \times 10^{-11}$; Fig. 4c).

Of the ten core chromosome regions targeted for deletion, the only region retained in the colony with nine deletions was between segments 6 and 7. The single gene residing in this region is *SYN8*, which encodes a SNARE protein involved in vesicle fusion with membranes[40]. Re-examination of the red colonies found that *SYN8* was retained in all 10, suggesting that it genetically interacts with one or more of the other deleted elements. To test for such an interaction, we deleted *SYN8* from the strain with the other deletions. In this context, deleting

*SYN8* increased doubling time to 394 min, an 86% increase over the multiple deletion strain and a 319% increase over the strain with a synthetic BY ChrI lacking the deletions (Fig. 4c; Supplementary Table 22). By contrast, *SYN8* deletion had no effect on BY (difference = 1 min; t-test, *p* value = 0.96; Fig. 4c; Supplementary Fig. 14).

The fact that we could generate the *SYN8* deletion in the presence of all other intended deletions suggests that we failed to recover this strain during initial screening due to its poor growth. In the initial iteration of the assembly, we picked colonies after native chromosome elimination, at which point cells containing all intended deletions would have been at a severe disadvantage due to their low fitness. To mitigate this issue, we redid the multiplex deletion assembly, this time picking individual colonies after chromosome assembly, rather than after native chromosome elimination. We then genotyped aneuploid colonies for the *SYN8* deletion and individually subjected aneuploid strains with the *SYN8* deletion to native chromosome elimination. Redoing the experiment in this manner led to the recovery of a strain

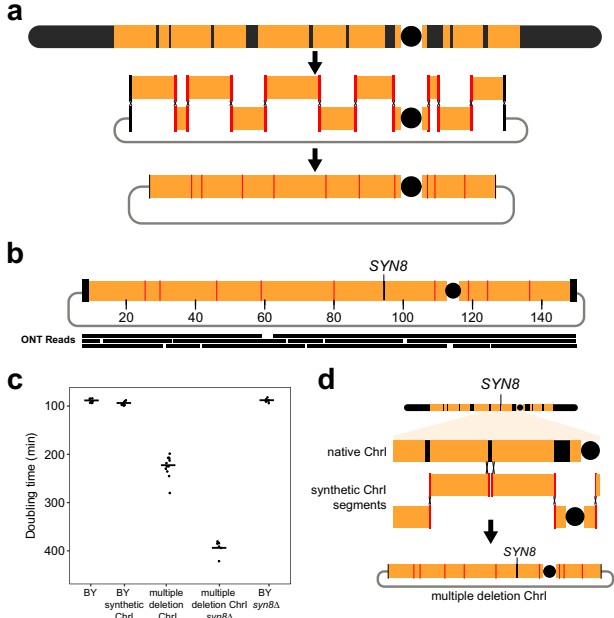

**Fig. 4 | Multiplex gene deletion with CReATiNG. a** We attempted to delete 10 non-adjacent regions of the chromosome core and both subtelomeres from BY ChrI, totaling 39.9% of the chromosome. To do this, we cloned 11 segments of the ChrI, assembled them, and performed native chromosome elimination. **b** ONT sequencing of a colony confirmed results from PCR checks. The colony with the most deletions (nine regions) had the correct structure, but had retained the region containing *SYN8*. An in silico design of the chromosome is shown with a subset of mapped reads plotted below it. **c** Growth rate analysis of different BY strains, including the unaltered reference strain (BY), a strain carrying a synthetic circular ChrI lacking subtelomeres (BY synthetic ChrI), a synthetic circular ChrI lacking nine core regions and both subtelomeres (multiple deletion ChrI), a synthetic circular ChrI lacking nine core regions, both subtelomeres, and *SYN8* (multiple deletion ChrI *syn8Δ*), and the reference strain with *SYN8* deleted (BY *syn8Δ*). For each strain, a total of nine independent growth experiments were performed (*n* = 9). Each dot represents the value from an individual replicate of a given genotype, while the horizontal bar represents the mean across the nine replicates. **d** Recombination between synthetic and native copies of ChrI produced synthetic chromosomes with *SYN8*.

containing all intended deletions, including *SYN8* (Supplementary Fig. 15). These results also show that segments of the synthetic chromosome can recombine with the native chromosome during assembly and prior to native chromosome elimination.

*SYN8* shows a different effect on growth when deleted at the same time as other chromosomal loci as opposed to when deleted individually. The explanation must be a genetic interaction with the multiplex deletion background, but the nature of this interaction is not clear. We looked at potential *SYN8* interactors among genes on ChrI using the Saccharomyces Genome Database[26,40]. There were zero genetic interactors, a single physical interactor at the mRNA level (Ccr4), and a single physical interactor at the protein level (Snc1). However, neither *CCR4* nor *SNC1* were eliminated in the multiplex deletion experiment. While it is conceivable that a change in their expression as a consequence of the multiplex deletion might produce a novel genetic interaction with *SYN8*, another possibility is that two or more genes that were simultaneously deleted interact with *SYN8* in a previously unidentified higher-order genetic interaction[41].

Our results show that CReATiNG can be used to eliminate many non-adjacent segments of a chromosome in a single assembly. However, they also highlight the difficulties that can be encountered in such experiments, as multiplex deletion can have substantial fitness consequences that make it harder to recover intended chromosome designs. In some cases, these fitness consequences can arise due to

unknown genetic interactions that may be important to identify. Our work shows that CReATiNG can also be used to detect such unknown interactions with genetic background, improving understanding of gene function and epistasis. In our study, multiplex deletion uncovered an unknown genetic interaction between *SYN8* and other elements on ChrI, which converted the normally dispensable *SYN8* into an quasi-essential gene[7]. Such unknown genetic interactions represent a major obstacle for efforts to streamline chromosomes and genomes, as they will cause strains carrying synthetic chromosomes to show slow growth and poor tractability in the lab. However, the *SYN8* example shows that synthetic chromosomes generated by CReATiNG can overcome such unknown interactions via recombination with native chromosomes prior to native chromosome elimination (Fig. 4d).

## Discussion

CReATiNG makes it possible to build synthetic chromosomes with diverse designs using natural components. Because CReATiNG employs cloned segments of natural chromosomes as opposed to small DNA fragments synthesized de novo, it is substantially cheaper and faster than de novo chromosome synthesis. For example, some of the final chromosomes completed for this paper went from in silico design to in vivo testing within a month and cost less than five hundred dollars to produce. Although some synthetic chromosome designs will require complete chromosome reprogramming, which is not possible with CReATiNG, many will not. Indeed, we have shown here that CReATiNG can be used to study a variety of fundamental questions in genetics, genomics, and evolution. Moreover, we unexpectedly found an additional benefit of CReATiNG, which is that it can allow cells to recover from unknown design flaws via recombination between a synthetic chromosome and its native counterpart.

While in vitro methods, such as ExoCET[42], CATCH[43], and CAPTURE[44], exist for cloning natural DNA segments, the in vivo cloning approach used in CReATiNG has advantages. It is highly efficient across a broad range of target sequences and sizes, and shows low sequence constraints relative to in vitro approaches. For example, some in vitro approaches for cloning natural DNA rely on Gibson assembly between a target segment and a vector, but such reactions depend on the overlap between the segment and the vector being within a few nucleotides of the end of the segment. With CReATiNG, the overlaps between a target segment and the homology arms in a vector do not need to be immediately at the end of a segment. This provides flexibility in where CRISPR cut sites can be located, as they do not need to be directly adjacent to homology arms. A potential constraint of cloning with CReATiNG is growth defects caused by genetic changes in donor yeast cells due to disruption of functional elements either by the programmed cutting or due to potential off-target effects. To minimize undesired phenotypic effects on donor cells, we designed all cloning reactions to avoid disruption of annotated functional elements. In this study, only one cloning reaction detrimentally impacted donor yeast cell growth. In this case, we were able to produce sufficient yeast cells for DNA extraction by growing the culture for an additional day, so there was no impact on our study.

Most of our work in this paper involved simple synthetic chromosome designs in which only three segments were assembled. However, we also demonstrated that CReATiNG can be used to build synthetic chromosomes with complex designs involving ≥10 segments. Using CReATiNG to make synthetic chromosomes with such complex designs could lead to important biological discoveries. For example, here we synthetically recombined chromosomes between strains and species using only two sites. However, this number could be increased, potentially by a large amount, facilitating fine scale genetic mapping of heritable traits within and between species. Similarly, more complex modifications of chromosome structure could be used to identify natural design principles governing chromosome

architecture. CReATiNG can also likely be used to delete larger numbers of linked, non-adjacent chromosome regions than explored here, facilitating rapid chromosome streamlining. In yeast, chromosome streamlining efforts are likely to encounter challenges associated with unknown genetic interactions that convert genes from non-essential to essential or quasi-essential, as in the *SYN8* example. Importantly, CReATiNG can be used to recombine synthetic chromosomes with native chromosomes to identify these problematic genes, ensuring they are not a barrier to progress.

In addition, CReATiNG can also be paired with de novo chromosome synthesis to enable projects that might not otherwise be feasible. For example, chromosomes with *Saccharomyces* architecture but sequences from other non-*Saccharomyces* species could be synthesized de novo and then recombined with *Saccharomyces* chromosomes. Such an experiment would facilitate study of the genetic basis of reproductive isolation and trait differences between phylogenetically distant organisms. In addition, CReATiNG and de novo chromosome synthesis could be employed in combination to efficiently relocate genes in the same pathways, complexes, or cellular processes to common genetic modules. Further yet, with some modifications, CReATiNG in yeast may be employed to produce or modify large DNA constructs, including synthetic chromosomes and gene variant libraries, for use in other systems, such as bacteria or mammalian cells. These diverse applications highlight how CReATiNG democratizes the use of chromosome synthesis in diverse biological research.

## Methods

### Production of target-specific cloning vectors

To produce the BAC/YAC cloning vector pASC1, we performed a four-piece Gibson assembly[45] with the pCC1BAC CopyControl plasmid from Epicentre Biotechnologies, a portion of the pRS316 plasmid[46] (ATCC #77145) that included *ARSH4*, *CEN6*, and *URA3*, and two DNA blocks containing I-SceI sites. To create the necessary homology for Gibson assembly, pCC1BAC and the portion of pRS316 were amplified by PCR with tailed primers. The two DNA blocks containing I-SceI sites were ordered from Twist Bioscience and also contain a multiple cloning site (MCS) that is used as a homology sequence between the two blocks. Correct assembly of all four pieces produced a vector with a multiple cloning site flanked by two I-SceI sites. To prepare this cloning vector for a specific target segment, we add a ~500 bp cloning cassette to pASC1 at the MCS. Each cloning cassette contains two ~150 bp homology arms flanked by ~100 bp adapters and separated by 30 bp of restriction sites that are used for vector linearization. Adapters are synthetic randomly generated DNA sequences of 100 bp in length and 40–50% GC content. These sequences were generated using an online random sequence generator tool (https://molbiotools.com/randomsequencegenerator.php). Cloning cassettes were ordered from Twist Bioscience. Addition of a cloning cassette to pASC1 was done by restriction digestion and ligation. Equimolar amounts of pASC1 and a cloning cassette were digested with EcoRI and SphI, and ligated using T4 DNA ligase. After addition of a cloning cassette, the vector was transformed into TransforMax EPI300 cells (LGC Biosearch Technologies) and high copy number was induced with Epicentre's CopyControl system via a copy control induction solution (LGC Biosearch Technologies). Large quantities of vectors were then harvested by ZymoPURE II plasmid midiprep kit (Zymo Research).

### In vitro transcription of gRNAs

The gRNAs were in silico designed using the built-in gRNA design tool present on the Benchling software (http://benchling.com). In summary, gRNAs with the highest predicted on-target scores, calculated by the software, within 1 kb of the homology sites used for cloning were selected as candidates. On-target score equal or higher than 75 was defined as a threshold to select the best gRNAs. When necessary, to avoid disrupting annotated functional elements this parameter was

relaxed to no lower than 50. The gRNAs for all CRISPR/Cas9 cutting experiments were produced by in vitro transcription[24]. For a given gRNA, we generated a dsDNA template by fusing two ssDNA oligonucleotides, one including the tracrRNA and the other the target-specific crRNA, using PCR. After PCR products were purified using the DNA Clean and Concentrator-5 kit, we combined 150 ng of purified PCR with 10 μl of RiboMAX 2X buffer and 2 μl of T7 express enzyme from the T7 RiboMAX Express Large Scale RNA Production System (Promega). Water was used to bring each reaction to a total volume of 20 μl. In vitro transcription reactions were incubated at 37 °C for ≥4 h. We then added 2 μl of DNAse and incubated reactions for an additional 18 min at 37 °C. We cleaned gRNAs using the RNA Clean & Concentrator-5 kit (Zymo Research) and stored them at −20 °C until use.

### Cloning of natural genomic segments

We clone target segments by co-transforming a linear version of pASC1 that possess homology arms matching the ends of a target segment, a repair template containing *KanMX*, and gRNAs into a *Saccharomyces* strain that constitutively expresses *Streptococcus pyogenes* Cas9 either from a chromosomally-encoded construct or from a plasmid. Prior to transformation, the cloning vector is linearized by cutting between the homology arms using AvrII and XhoI. The repair template is produced through a PCR reaction using a modified pRS316 plasmid in which *URA3* was replaced with *KanMX* as the template. The primers are designed to flank *KanMX* and contain 40 bp homology tails that match genomic sites adjacent to a target segment. Yeast cells were transformed with 200 ng of linearized vector, 200 ng of repair template, and 1 μg total of a mix of multiple gRNAs. Typically, we included six distinct gRNAs, three targeting each side of a segment. Cells were transferred to 2 mm electroporation cuvettes and electroporated at 2.5 kV, 200 Ω, and 25 μF[47]. Transformants were recovered for 2–3 h in YPDS, a 50:50 mix of YPD (2% glucose, 1% yeast extract, and 2% peptone) and 1 M sorbitol, and plated on SC Ura- plates containing G418 to select for the pASC1 vector and use of the repair template, respectively. After 2 days, transformants were checked by colony PCR at both junctions between pASC1 and a cloned segment.

### Amplification of cloned segments

Cloned segments were extracted from yeast using the Zymo Research ZymoPURE II bacterial midiprep kit and the following steps. 50 ml of yeast cells from an overnight culture were washed with water and centrifuged at $3000 \times g$ for 5 min. The cells were resuspended in Y1 buffer (1 M Sorbitol, 100 mM EDTA pH 8.0, 14 mM beta mercaptoethanol) and 1000 units of lyticase. Cells were incubated at 30 °C for at least 15 min. The spheroplast preparation was visually inspected every 15 min using a basic light microscope. Spheroplast cells are larger and more rounded than regular yeast cells. Reactions were stopped when the majority of analyzed cells appeared to be spheroplasts. Typically, this took between 45 min and 1 h. The spheroplasts were pelleted and washed with sterile water twice. The remaining steps proceeded according to manufacturer recommendations. To prevent DNA shearing, vortex steps were avoided and wide bore pipette tips were employed. The plasmid containing the cloned segment was then transformed into EPI300 cells (LGC Biosearch Technologies) and transformants were selected by growing in LB chloramphenicol plates (30 μg/ml). Presence of the correct cloned segment was checked by junction PCRs and confirmed transformants were grown overnight at 37 °C on a shaking incubator. High copy plasmid induction was done by transferring 4.5 ml of overnight growing cells to 45 ml of LB supplemented with chloramphenicol (30 μg/ml) and 50 μl of copy control induction solution (LGC Biosearch Technologies). Cells were grown for 5 h and plasmids extracted using the midiprep kit described before, using manufacturer recommendation. The cloned segments were liberated from the cloning vector by digestion with I-SceI, followed by 0.5% agarose gel electrophoresis at 70 V for 90 min, and purification

with the Zymoclean Large Fragment DNA Recovery kit (Zymo Research).

## Construction of the centromere cassette

We generated the centromere cassette by adding *KanMX* and a loxP site to the pRS316 plasmid right after its *CEN6/ARS4* region. The pRS316 vector, the *KanMX* cassette, and a loxP site were amplified using primers with homology tails. The molecules were mixed in equimolar amounts and ligated using Gibson Assembly Master Mix (New England Biolabs). These assemblies were then transformed into Dh5α cells, and the correct assembly was identified by PCR and Sanger sequencing. To generate the centromere cassettes used in synthetic chromosome assemblies, we amplified the centromere cassette region of the plasmid with tailed primers containing appropriate adapters.

## Chromosome assembly

Synthetic chromosomes were assembled as circular molecules including segments, a centromere cassette with appropriate adapters, and a modified pASC1 vector named pASC2, which lacks CEN/ARS and contains *HIS3* instead of *URA3*. A given assembly was performed by co-transforming 500 ng of each purified segment, 200 ng of the centromere cassette, and 200 ng of linearized pASC2 into BY. Transformation was performed using a standard PEG/LiAc method[48], but extra care was taken when handling DNA solutions to avoid DNA shearing. All vortex steps were replaced by gentle manual shaking and pipetting with wide bore tips. Transformants containing assemblies were selected on SC lacking histidine and containing G418. Correct assemblies were identified by junction PCRs and confirmed by ONT sequencing.

## Elimination of native chromosomes

We used CRISPR/Cas9 to place *pGAL1* 114 bp upstream of the centromere on the native ChrI in BY. In the presence of galactose, *pGAL1* drives transcription through the centromere, destabilizing its function and resulting in native ChrI loss in some cells[12,29]. We also marked the native ChrI with *URA3*, enabling selection for native ChrI loss. After assembly, cells possessing an assembly were selected on SC lacking histidine and containing G418, and then replicated or individually streaked out onto plates that lacked histidine and contained galactose. Cells were grown for two days and then replicated or individually streaked out into SC plates lacking histidine that were supplemented with 5-FOA. Elimination of native ChrI was confirmed by diagnostic PCRs of sites that present on the native ChrI but not a synthetic ChrI.

## Linearization of synthetic *S. paradoxus* ChrI

The BY strain containing the assembled *S. paradoxus* ChrI was first transformed with a modified version of the pML104 Cas9 plasmid[23], which had *URA3* replaced with *HIS3* (Supplementary Table 19). To linearize the *S. paradoxus* ChrI, we used electroporation to co-transform two telomere cassettes and two pairs of in vitro transcribed gRNAs that target the ends of the pASC2 vector (Supplementary Table 20). The left telomere cassette was generated by amplification of *URA3* in the pRS316 vector using primers with tails that added a telomere seed sequence and homology to *S. paradoxus* ChrI (**For:** TGTGTGGTGTGTGGTGTGTGTGGGTGTGTGGTGTGTGGGTCTG TAAGCGGATGCCGG;**Rev:**CTCCTTACGCATCTGTGCGTACCCTTTAA AATCTCATTGGCTCGTGATTAATTTGTTCTGTGCTGCTGAATATTCA TGC). The right telomere cassette was generated by amplifying *NatMX* from a modified pRS316 plasmid, which had *URA3* replaced with *NatMX* (**For:** TTACATATCCTCTACACCGAGCGCGTCGACCCGTCGA ATGGTTTAGCTTGCCTTGTCCCC;**Rev:**GGCGGCGTTAGTATCGAATC CACCCACCACACACACCCACACACACCACACACCCACCCA). Linearization occurs when the gRNAs create double strand breaks at both ends of pASC2, which are repaired by homologous recombination between *S. paradoxus* ChrI and the left and right telomere cassettes. Strains with a linear ChrI will have *URA3* and *NatMX*, and will lack the

*HIS3* on pASC2. Linearization was confirmed phenotypically and by PCR of the linearization sites.

## Growth assays

Most phenotyping was done using liquid growth assays on a BioTek ELx808TM 96-well plate reader. A given strain was grown overnight in YPD at 30 °C and 1.2 µl was then inoculated into 118.8 µl of YPD in the appropriate wells of a 96-well plate. A randomized block design was used to mitigate positional effects. The plates were incubated with shaking at 30 °C or 35 °C on the plate reader and OD600 was acquired every 15 min until cultures reached the stationary phase. Doubling time value for each culture was calculated using PRECOG (PREsentation and Characterization Of Growth-data)[49]. We also performed dilution spot assays. Overnight cultures and their ODs were measured. Cell aliquots were diluted in YPD to an OD of 1. We then performed tenfold serial dilutions (1:10, 1:100, 1:1000, 1:10,000). 3 µl of each dilution, including non-diluted overnight culture, were pinned into appropriate plates and incubated at 30°C or 35°C. Plates were imaged using a BioRAD Gel Doc XR+ Molecular Imager at a standard size of $11.4 \times 8.52 \, cm^2$ (width × length) and imaged with epi-illumination using an exposure time of 0.5 s. Images were saved as 600 dpi tiffs.

## Quantitative analysis of the effect of each segment on growth at 35 °C

To measure the additive and epistatic effects of the 3 ChrI segments among the 27 chimera strains, growing at 35 °C we implemented full factorial ANOVAs in R. Specifically, we modeled the median doubling time of the chimera strains as a function of all possible additive and epistatic effects involving the three segments. The model was specified using the statement: lm(Temperature_35C ~ Segment1 * Segment2 * Segment3). ANOVA tables were then obtained using the anova() function. In addition to the terms provided by R, we computed the percent of phenotypic variance explained for each segment by dividing the sum of squares associated with a particular term by the sum of squares total (Supplementary Table 9). Respectively, the fractions of phenotypic variance explained by all genetic effects or only additive genetic effects were computed by summing the fractions of phenotypic variance explained by all genetic terms or only additive genetic terms in a given model.

## RNA extraction for RT-qPCR analysis

Four biological replicas of each strain IEY402, 421 and 425 were cultivated overnight and individually transferred to 5 ml YPD at a starting OD of 0.2. Cells were cultivated at 30 °C with 200 rpm agitation and harvested when OD reached 0.8. Cells were washed twice with 5 ml of water. RNA extraction was then performed using the YeaStar RNA extraction kit (Zymo Research) following the manufacturer's protocol.

## RT-qPCR analysis of *MYO4*, *SNC1* and *NUP60*

Primers were designed using the PrimerQuest tool (IDT integrated DNA technologies) using the design option "2 primers qPCR intercalating dye". All RT-qPCR reactions were performed on AriaMx Real-Time qPCR System (Agilent Technologies), using the Luna Universal One-Step RT-qPCR kit (New England Biolabs Inc.). The standard curves for the four genes were constructed by fourfold serial dilutions and analyzed using the High-Resolution Melt qPCR software (Agilent Technologies). The linear correlation coefficients obtained from each standard curve ranged from R: 0.96–0.99.and the calculated primer amplification efficiencies (*E*) were 109%, 97%, 110% and 108% for *MYO4*, *SNC1*, *NUP60* and *ACT1*, respectively. Both linear correction and primer efficiency were automatically calculated by the software. For each RT-qPCR experiment, two different negative controls were used: one containing all reaction components except cDNA and another containing all reaction components except primers. RT-qPCR amplification was conducted in optical-grade 96-well plates (Agilent

Technologies) and 1 l of RNA per sample was added to the reaction solution in accordance with the Luna Universal One-Step RT-qPCR kit's protocol. Each reaction was performed in triplicate. The reaction conditions were an initial step at 55 °C for 10 min followed by 95 °C for 1 min and then 40 cycles of 95 °C for 10 s and 60 °C for 30 s. A melting curve step was also performed according to the AriaMx Real Time qPCR System's recommendation, using 95 °C for 30 s, a 65–95 °C degree ramp with 0.5 °C increment and a soak time of 10 s, and 95 °C for 30 s. The Ct was determined automatically by the instrument and gene expression ratios were calculated using the Pfaffl method[50]. *ACT1* was used as a control housekeeping gene and IEY402 strain (1-2-3) was the control strain. For each analyzed gene in each strain other than IEY402, mean values of gene expression ratios were compared against the gene expression ratio for that gene in IEY402 using t-tests and a significance threshold of $p \le 0.05$.

### Deletion of *SYN8*
We used Cas9 to delete *SYN8* from BY and the ChrI multiple deletion strain. Both strains were electroporated with the pML104 Cas9 vector that had *URA3* replaced with *HIS3*. We used a 2 mm cuvette and 2.5 kV, 200 Ω, and 25 μF. Transformants were selected on SC His-. We then performed a second electroporation with 200 ng of a *NatMX* repair template with homology arms targeting the sites adjacent to the cut and 1 μg total of gRNAs targeting *SYN8* (Supplementary Table 19). We added the homology arms to *NatMX* by tailed PCR using the modified pRS316 vector as the template (**For**:GGGCTATAAAGTATATA TAGATACAAATATATGATGAATCGTTTAGCTTGCCTTGTCCCC;**Rev**: GAATAAAATTTCCCAGCACGACTTTGATCACCCGAAAGGGGGCGGCG TTAGTATCGAATC). Transformants were recovered for 2-3 h in YPDS and plated on YPD plates containing 200 ug/ul of nourseothricin (Goldbio, USA).

### Genomic DNA extraction
Strains were streaked from a −80 °C freezer stock onto YPD plates containing G418 and allowed to grow for 2 days at 30 °C. A single colony was then used to inoculate a 5 ml overnight culture in YPD containing G418, which was placed in a shaker at 30 °C. The 5 ml overnight culture was then inoculated into 50 ml +G418 in a 250 ml Erlenmeyer flask. The cells were shaken overnight at 30 °C. Prior to extraction, these cultures were normalized to $7.0 \times 10^9$ number of cells in YPD. Cells were then placed into 50 ml falcon tubes, centrifuged at $3000 \times g$ for 10 min at 4 °C, and supernatant was decanted. Cell pellets were gently washed with 10 ml of PBS and centrifuged again at $3000 \times g$ for 10 min at 4 °C. Supernatant was then decanted and a fresh 10 mL of PBS was added. This process was repeated twice. After the wash steps 4 ml of Y1 (1 M sorbitol, 100 mM EDTA pH 8.0, and 14 mM beta mercaptoethanol which was added immediately before use) were added to the cells along with 1 ml of lyticase (15 U/μl). The tubes were transferred back to the shaking incubator at 30 °C for ≥1 h, with hand inversion done every 10 min to prevent the cells from settling. OD was checked periodically and once the cells had lost 80% of their OD660, spheroplasting was considered complete. Spheroplasts were centrifuged at $5000 \times g$ for 10 min at 4 °C, and the supernatant was decanted and replaced with 5 ml of Qiagen G2 buffer with 15 μl of RNase A and 300 μl of Proteinase K. The solution was incubated at 55 °C for ≥2 h. A Qiagen 100/G Genomic tip was equalized with 4 mL of QBT buffer. While this was flowing through the samples were centrifuged at $5000 \times g$ for 10 min at 4 °C to spin down cell debris. The sample was passed through the column, collected and repassed through the column a second time. The column was washed with $2 \times 7.5$ ml of buffer QC, then eluted with 5 ml of buffer QF. The collected elution was passed through the column a second time, then 3.5 ml of molecular grade isopropanol was added along with 0.01× volumes of filter sterilized sodium acetate. The samples were placed in a −20 °C to precipitate for 48 h. After precipitation, the samples were spun down at

$21,000 \times g$ for 10 min at 4 °C to collect the DNA in a Lo-bind tube. The ethanol was decanted and the ethanol was allowed to evaporate for 1 min. An extraction buffer (EB) of 1080 μl of 10 mM tris-HCl, 1 mM EDTA Ph 9.0, and 45 μl Triton X-100 .5% (v/v) was created fresh before each use. 200 μl of EB buffer was added to the DNA. The DNA was then allowed to dissolve at 55 °C for 30 min followed by 2–3 h at 30 °C.

### Oxford Nanopore Technologies library preparation
For ONT library preparation, the SQK-LSK109 and EXP-NBD104 protocols were used with the following modifications. At the end of each bead clean up, DNA was eluted in EB at 37 °C for 10 min to allow for the recovery of larger molecules. The library was then barcoded and multiplexed with up to ten samples. These multiplexes were sequenced on R9.4.1 Chemistry, MIN-106 flow cells. The.fast5 archive was Tarbelled, gunzipped, and uploaded onto the USC's Center for Advanced Research Computing (CARC) Discovery cluster. Basecalling was performed using Guppy (v. 6.0.1), which was contained in a Singularity container using the configuration dna_r9.4.1_450bps_sup.cfg with 16 threads and a V100 GPU. The.fastq files were demultiplexed and trimmed of barcodes in Guppy using standard parameters.

### Sequence analysis of synthetic chromosomes
For each strain carrying a synthetic chromosome, we built a reference genome in silico using appropriate data from BY, RM, or *S. paradoxus* CBS5829. Because all assemblies were performed in BY, Chromosomes II-XIV, as well as the mitochondrial genome and 2-μm plasmid, were always taken from the BY reference genome. The only part of the reference genomes that varied was ChrI. Reads were mapped against the appropriate reference using minimap2[51] (v. 2.24-r1122, -ax mapont). Using samtools[52,53] (v1.15, htslib v1.15, view -bS), the.sam was converted into a.bam file, sorted, and indexed. We then used bamtools (v2.5.2) to split the.bam file by chromosome. The.bam file for ChrI was extracted and used in both Nanocaller[54] and Sniffles[55] with variant calls being made when both programs agreed upon a variant and the.bam files could be visually inspected to verify each call. Reads spanning adapters were extracted using samtools (v1.15, htslib v1.15, view) and checked by visual inspection. We required a minimum average per site coverage of 10× for structural variant calling and 21x for SNP calling due to the high error rate associated with ONT reads. In some cases, a strain had to be sequenced on a second subsequent flow cell because its initial coverage was insufficient in its initial multiplex sequencing run. Coverage per site was calculated using the depth function in samtools (v1.15, htslib v1.15, -a).

### Reporting summary
Further information on research design is available in the Nature Portfolio Reporting Summary linked to this article.

## Data availability
All data generated or analyzed in this study are included with this manuscript via the figshare link https://doi.org/10.6084/m9.figshare.24251383. Raw sequencing data is available through the NCBI Sequence Read archive under the accession number PRJNA986901. Source data are provided with this paper via the above figshare link. Plasmids are available via Addgene or by request. Source data are provided in this paper.

## Code availability
Analyses were conducted using off-the-shelf software and methods. Code for running software and plotting data is available via the figshare link https://doi.org/10.6084/m9.figshare.24251383.

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

## Acknowledgements

We thank Oscar Aparicio and Steven Finkel for allowing us to use equipment in their labs and Norman Arnheim, Steven Finkel, Joseph Hale, Julia Schwartzman, and John Tower for feedback on a manuscript draft. This work was supported by startup funds from the University of Southern California, grants 2124400 from the National Science Foundation and R35GM130381 from the National Institutes of Health to I.M.E., as well as an Agilent Postdoctoral Fellowship to A.L.V.C.

## Author contributions

A.L.V.C. and I.M.E. conceptualized this project. A.L.V.C., C.N.V., Z.A.K., J.R., C.H., S.Y. and D.T.L. performed the experiments. A.L.V.C. and C.N.V. analyzed the data. A.L.V.C., C.N.V. and Z.A.K. generated the figures. A.L.V.C. and I.M.E. wrote and edited the manuscript, with substantial input from C.N.V. and Z.A.K.

## Competing interests

The authors declare the following competing interest: the University of Southern California has filed a non-provisional patent application (530.029WO1) with A.L.V.C. and I.M.E. as named inventors covering the entire process of cloning natural DNA segments and assembling these segments into chromosomes inside living cells disclosed in this manuscript. All other authors declare no competing interests.
