## [Peer Review File · Nature Communications]

Reviewers' Comments:

Reviewer #1:

Remarks to the Author:

The paper by Coradini, Ne Ville et al. presents CReATing - a method for constructing synthetic chromosomes from natural DNA. CReATing is based on CRISPR-Cas9 excision of segments of native chromosomes and capturing them on a BAC/YAC by in vivo homologous recombination. Following verification of the assembly and amplification in *E. coli*, segments are transformed into the destination yeast strain and assembled in vivo using recombination of unique adapters. As a proof of concept, authors construct chromosome I from *Saccharomyces paradoxus* in *S. cerevisiae* and further expand it by swapping positions of chromosome segments and constructing a library of 27 version of synthetic chromosome I using chromosome segments originating from three different strains. Finally, authors show that CReATing can be used to construct a synthetic chromosome from as many as 11 segments which allows for multiplex gene deletion.

The CReATing method is essentially a new workflow that comes from linking together a set of existing methods that individually have been used lots of times before by others working in yeast genomics and synthetic genomes (e.g. in the Sc2.0 consortium or NYU Dark Matter Project). The novelty here is not really in the methods or the work, but in the concept. By linking together existing steps into a workflow, the authors can now present a more streamlined, dedicated approach to assembling chromosome-scale constructs in yeast from diverse DNA sources. This is certainly likely to be of value and interest to some of your journal readership, and the final story in the paper about SYN8 is quite interesting too.

Throughout the paper, the experiments were well designed and data in both the main text and supplementary information are clearly presented. The main drawback of the paper is lack of some essential details that help readers understand the workflow (listed below).

Key comments

1. The main text doesn't contain any methods section which is crucial considering that this is a methods paper. Please ensure that the revised version has the key methods moved to the main text and not all hidden in the supplementary sections.
2. The authors used a CRISPR-Cas9 system to capture a segment of chromosome but in the main text it is not specified how Cas9 is expressed in these cells, other than it is constitutive. It would be interesting to know whether Cas9 was expressed from a plasmid (and if so what type of plasmid) and or if it is a genome integrated version, and if so at what locus. What promoter is used to express Cas9?
3. Page 6 - In this method a region of natural chromosome is cloned into the capture vector pASC1. Can the authors clarify if this step affects growth and fitness of yeast, and whether this could be a constraint for culturing the generated strain to extract the YAC (which would require high amounts of cells).
4. Page 6 - cloning a segment of chromosome into the capture vector pASC1 is only verified by junction PCR. Is this sufficient? Surely this approach would not detect if any chromosomal segments contain a deletion, and might also not detect duplications either. The same issues with junction PCR arises for page 9 (third paragraph). A long-read sequencing approach would be much more desirable here.

Minor comments

1. The paper will be easier to follow if the authors could name all the plasmids and strains, list them in a supplementary table and refer to them in text.
2. Page 3 - please also include the number of genes in *Mycoplasma mycoides* such that the reader can understand how many genes were reduced in the synthetic chromosome.

3. Page 8 – Is there any explanation as to why segment 3 has the most pronounced impact on the growth at higher temperature?
4. Figure 2B and C – under both growth conditions the three top performing strains are consistent. Do the authors have any hypotheses on why these combinations are improving the performance compared to the wild type BY and RM?
5. Figure 2B and C – is the number of experimental replicates missing here (is it 12?)
6. Figure 3B – the strain with segments 1-3-2 has twice the length of doubling time. Do authors have any explanation for this particular orientation causing a growth defect?
7. Methods, Cloning of natural genomic segments – can authors elaborate on how three gRNAs were designed to cut each site of the segment? What would be the distance between the 3 generated cuts? How does it affect the length of the region used for homologous recombination if some gRNAs are not functional?
8. Methods, Amplification of cloned segments – “inspecting spheroplasts every 15 min” – how was this done? Please explain more?
9. Methods, Chromosome assembly – “co-transforming >500 ng” – can you please describe how the amount of DNA of each segment was determined?
10. Methods ONT library prep – authors claim multiplexing up to 10 strains for the whole genome sequencing on MinION. What was the minimal depth of the sequencing reads for each strain in this multiplex experiment?
11. Supplementary Figure 8 – this figure will be more readable if ONT reads are not included and the figure only presents the combinations of segments.
12. Supplementary Table 1 and 19 – It would be more easy for readers if annotated sequences of plasmids were provided (e.g. Genbank files or Benchling links) rather than just giving a sequence in table.
13. Why did the authors decide to transform in vitro transcribed gRNAs rather than express gRNAs in vivo? This is not a common choice for yeast research.

Reviewer #2:

Remarks to the Author:

Coradini et al. described a CRISPR-Cas9 assisted gap-repair cloning strategy that efficiently clones native yeast chromosome fragments ranging from 3.8 kb to 64 kb. Thanks to the unique flanking adaptors, the authors were able to assemble the cloned fragments into a larger construct in a defined order in the subsequent step. Taking advantage of this flexibility, the authors first assembled 27 possible Chr.I recombinants consisting of three fragments cloned from three *Saccharomyces* strain/species in their native order. Second, the authors reorganized the fragment orders and assembled 5 non-natural Chr.I structures using the 3 intermediate Chr.I fragments cloned from the *S.cerevisiae* BY strain. In both cases, the doubling time was measured as a readout. Lastly, the authors intended to delete 10 regions in the genome using their method, which required cloning 11 intermediate fragments, then seemed combined the assembly step with native Chr.I elimination step in one transformation, but failed to recover colonies with all 10 deletions. After examining a clone with 9 deletions, they found a potential important genetic interaction between SYN8 and other deleted genes. This manuscript provides a cost-effective way to “shuffle” a native yeast chromosome by design, but the applications of this technique seem somewhat limited. These are the major concerns to be addressed:

1. The authors cloned yeast chromosome I as a circular BAC/YAC plasmid, but they did not determine the plasmid copy number. A plasmid copy number quantification assay should be

performed before analyzing the strain fitness. This is important because the CEN/ARS sequences seem to lead to higher than 1 copy/cell e.g. in the pRS vector series. Ideally, comparison can be made to the native copy number before the chromosome destabilization step.

2. In Figure 2 and 3, where are the boundaries of the three fragments, how they are defined? Is it possible that reorganizing might simply have altered the gene expression near the fragment boundaries, RNA profiling can help sort this out?

3. In Figure 4, the adaptor sequences are left behind in the genome, it seems the author could make the deletion scarless by using the next fragment's sequence as adaptor – why are the special adaptors even needed? Also, if the authors want to make the multi-gene deletion strategy generalizable, they should design a selection strategy that allows replacement of the native chromosome segment with the assembled segment. In this manuscript the authors relied on the deletion of ADE1, which is not an efficient way to screen for the desired construct. Wouldn't a simple drug resistance marker or the like be much more efficient?

4. The annotation of the SYN8 gene on SGD claims that the null mutation has increased competitive fitness, which contradicts what this manuscript reports, the authors should provide some thoughts on why their conclusion is contradictory. Also, the authors could search previous genetic interaction studies to see if there are known important interactions between SYN8 and other genes on Chr. 1.

5. Title : While CREATE is a cute name, this isn't creating a sequence from scratch; Better to use "recombinant chromosome" instead of "synthetic chromosome" to describe the strains they generated.

Reviewer #3:

Remarks to the Author:

In this manuscript, the authors reported a method for constructing synthetic chromosomes from natural components in yeast, called CReATiNG. They demonstrated the approach by native chromosome replacement, chromosome structure modification and multiplex gene deletion. The advantage of the approach is its feasibility and convenient operation to obtain a long segment of natural DNA when compared with PCR-based or DNA synthesis-based methods.

However, my major concern is that similar studies have achieved cloning of natural segment, such as CATCH, reported by Wenjun Jiang et al. and CAPTURE, reported by Behnam Enghiad et al. The authors should describe the differences or improvements between CReATiNG and these two methods. In addition, the method is highly depended on the ability of homologous recombination of the host, it is hard to be applied in other organisms.

The other major concern I have is about the approach. Is that the recombination efficiency would decrease rapidly with increased sizes and number of segments. Are the three segments used for the chromosome assembly with the same selection marker? The segments were released from the plasmid by I-SceI in vitro. The following steps were the same with standard TAR-based assembly. Is it possible to finish these steps in vivo? Because when the size of segment increases, it is more difficult to extract and purify the required segments from the yeast cell.

Other minor issues:

1. In Figs. 2e and 2f, the variations within the groups were much larger than those between them. I don't understand what the dots represent in each group. Was one dot represented one strain with a recombinant chromosome? It was rather imprudent to get the conclusion that "all three ChrI segments contribute to growth variation across temperatures". The conclusion should be more specific after additional validate experiments.

2. Whether the gene compositions and organizations of the three segments from three species were different as they showed different length (Fig. S7). If there were differences, it may be related to the phenotype variations.

3. It was interesting that the 1-3-2 configurations showed growth reductions compared to the natural 1-2-3 configuration. More experiments such as gene expression profile in these strains

could be provided to explain this growth deflection.

4. I do not think the last application of CReATiNG was a good example as its performance was more complicate than PCR-based assembly. In addition, CReATiNG cannot provide a solution for highly multiplexed deletion as the designs for all the segments were knowledge-based. The success of deletion of SYN8 in the multiple deletion strain also proved that it was possible to construct the targeted strain with all sequence deletion. However, the wild-type copy of Chr I hindered the synthetic chromosomes generation.

October 8, 2023

To the Reviewers,

Thank you for your constructive feedback on our manuscript. All review criteria have been met:

- All sequencing data are now available via the NCBI Sequencing Read Archive Project PRJNA986901.
- All scripts for data processing are now provided via Figshare and documented under Code Availability, following the Methods. Analyses were performed using off-the-shelf software, as discussed in the Methods.
- All data underlying figures were already included in the Supplement.
- No image manipulation was performed on gels. The relevant portion of gels are uncropped and unmanipulated. In line with Nature Communications requirements, all bands are shown.
- We have modified Figures 3b and 4c to show underlying data points.

We have provided a point-by-point response to all reviewer comments below. To simplify the checking of revisions, we have submitted two copies of the revised manuscript: one that has alterations to the main text and supplement highlighted and one that does not. The version with highlighting is supplied as a related manuscript file. Also, we note that due to modifications to the paper, we changed the 'Results' and 'Discussion' sections to the 'Results and Discussion' and 'Conclusion' sections, respectively.

Sincerely,

Alessandro Coradini and Ian Ehrenreich

REVIEWER COMMENTS

Reviewer #1 (Remarks to the Author):

The paper by Coradini, Ne Ville et al. presents CReATing - a method for constructing synthetic chromosomes from natural DNA. CReATing is based on CRISPR-Cas9 excision of segments of native chromosomes and capturing them on a BAC/YAC by in vivo homologous recombination. Following verification of the assembly and amplification in *E. coli*, segments are transformed into the destination yeast strain and assembled in vivo using recombination of unique adapters. As a proof of concept, authors construct chromosome I from *Saccharomyces paradoxus* in *S. cerevisiae* and further expand it by swapping positions of chromosome segments and constructing a library of 27 version of synthetic chromosome I using chromosome segments originating from three different strains. Finally, authors show that CReATing can be used to construct a synthetic chromosome from as many as 11 segments which allows for multiplex gene deletion.

The CReATing method is essentially a new workflow that comes from linking together a set of existing methods that individually have been used lots of times before by others working in yeast genomics and synthetic genomes (e.g. in the Sc2.0 consortium or NYU Dark Matter Project). The novelty here is not really in the methods or the work, but in the concept. By linking together existing steps into a workflow, the authors can now present a more streamlined, dedicated approach to assembling chromosome-scale constructs in yeast from diverse DNA sources. This is certainly likely to be of value and interest to some of your journal readership, and the final story in the paper about SYN8 is quite interesting too.

Thank you. We agree with many of these comments, though we would add that the cloning method enabling terminal reprogramming of genomic segments with adapters or other sequences is novel. There are limited options for reprogramming the termini of natural DNA molecules, but doing so is necessary to utilize natural DNA as the substrate for producing non-natural chromosome designs.

Throughout the paper, the experiments were well designed and data in both the main text and supplementary information are clearly presented. The main drawback of the paper is lack of some essential details that help readers understand the workflow (listed below).

In response to this comment and as discussed below, we have added more detail to the Results and Discussion, especially in the sections: 'A system for cloning and reprogramming natural DNA for assembly', 'Initial assembly of a chromosome using CReATiNG', and 'Recombination of chromosomes between strains and species'.

Key comments

1. The main text doesn't contain any methods section which is crucial considering that this is a methods paper. Please ensure that the revised version has the key methods moved to the main text and not all hidden in the supplementary sections.

As mentioned above, we have incorporated substantially more methodological details into the Results and Discussion, particularly the sections 'A system for cloning and reprogramming natural DNA for assembly', 'Initial assembly of a chromosome using CReATiNG', and 'Recombination of chromosomes between strains and species'. Details that are now more explicitly discussed in the main text include how the cloning process works, including key reagents such as Cas9 and gRNAs, and how certain design choices were made, including how random sequences for DNA adapters were determined and rationale for dividing ChrI into three segments in the 'Recombination of chromosomes between strains and species' section. See the sections and lines below for changes:

- Section 'Results and Discussion': pages 5 and 6, lines 96 -130 and 135-142.
- Section 'Initial assembly of a chromosome using CReATiNG': page 7, lines 147-150; 153-161;164-169.
- page 8, 178-186; 196-197.

2. The authors used a CRISPR-Cas9 system to capture a segment of chromosome but in the main text it is not specified how Cas9 is expressed in these cells, other than it is constitutive. It would be interesting to know whether Cas9 was expressed from a plasmid (and if so what type of plasmid) and or if it is a genome integrated version, and if so at what locus. What promoter is used to express Cas9?

We have added substantial details along these lines to the section 'A system for cloning and reprogramming natural DNA for assembly'. In this study, Cas9 was constitutively expressed from a strong constitutive promoter (*TDH3*) off a plasmid (pML104) carrying the *HIS3* marker. We transformed pML104 into a donor yeast strain prior to attempting any cloning reactions. We used in vitro transcribed (IVT) gRNAs instead of more conventional gRNA plasmids because the former allows PCR products to be used as templates, eliminating assembly and verification of gRNA plasmids. IVT gRNAs also make it possible to use several distinct gRNAs for the same target in the same transformation with little added work, removing the need to screen individual gRNAs for efficacy. We employed three gRNAs for each side of a target segment (six total per segment), choosing gRNAs with the highest predicted on-target scores within 1 kb of the homology sites used for cloning. If genes were present within the 1 kb regions, preference was given to gRNAs that had lower on-target scores

but were outside known coding or regulatory regions. The maximum distance between gRNAs targeting the same side of a segment was ~200 bp. See the section and lines below for changes:

- Section 'Results and Discussion': pages 5 and 6, lines 96 -115.

3. Page 6 - In this method a region of natural chromosome is cloned into the capture vector pASC1. Can the authors clarify if this step affects growth and fitness of yeast, and whether this could be a constraint for culturing the generated strain to extract the YAC (which would require high amounts of cells).

Among all cloned segments in the paper, we found only one affected growth to a degree that impacted subsequent steps. We were able to mitigate this issue by simply growing the donor yeast culture for an extra day. We mention this in the Conclusion, noting it as a potential concern, and also provide more details about BAC/YAC extraction in the Methods. It could be that the reason this was not an issue in our study is that we designed cloning reactions in a way that was informed by annotated functional elements and should not have disrupted any known essential genes. See the sections and lines below for changes:

- Section 'Conclusion': pages 17 and 18, lines 451-458.
- Section 'Amplification of cloned segments' from Methods: page 4, lines 105-113.

4. Page 6 – cloning a segment of chromosome into the capture vector pASC1 is only verified by junction PCR. Is this sufficient? Surely this approach would not detect if any chromosomal segments contain a deletion, and might also not detect duplications either. The same issues with junction PCR arises for page 9 (third paragraph). A long-read sequencing approach would be much more desirable here.

Checking junctions by PCR and Sanger sequencing and checking inserts by restriction digestion and gel imaging alone might seem insufficient. However, we checked all chromosomes that were assembled by long read sequencing, most at high enough coverage to confirm not only structures but also nucleotide sequences. No instances of large deletions or duplications were observed. Generally there were few mutations and observed mutations were largely the result of using PCR to generate components, such as the centromere cassette. Part of the reason we proceeded in this manner was that in our initial work to build a *S. paradoxus* chromosome inside *S. cerevisiae*, we did not find any evidence that our cloning and amplification process for BAC/YACs exhibits detectable mutagenicity.

Minor comments

1. The paper will be easier to follow if the authors could name all the plasmids and strains, list them in a supplementary table and refer to them in text.

We have added the Supplementary Tables 1 and 2 that contain all the plasmids and strains from this study and now mention the plasmids by name in the main text. We have also deposited key plasmids into Addgene. Strains are now referenced in the main text according to their IDs as shown on Supplementary Table 2.

2. Page 3 – please also include the number of genes in *Mycoplasma mycoides* such that the reader can understand how many genes were reduced in the synthetic chromosome.

We added the following sentence to the Introduction: 'Generating this minimal cell involved eliminating 428 (48%) of the genes that are naturally present in *M. mycoides*.' See page 3, lines 39 and 40.

3. Page 8 – Is there any explanation as to why segment 3 has the most pronounced impact on the growth at higher temperature?

Thermotolerance between *S. cerevisiae* and *S. paradoxus* shows very high genetic complexity, with >50 contributing genes based on recent work. Such quantitative trait loci can show a broad range of effect sizes, consistent with what we have seen here. Unfortunately our mapping resolution was coarse because we split the chromosome into only three pieces, limiting our ability to identify causative factors in segment 3. In response to this comment and a comment from Reviewer 3 comment 3, we added some text discussing gene content differences between *S. cerevisiae* and *S. paradoxus* in segment 3. However, determining whether these gene content differences or other genes are responsible for the effect of segment 3 is a significant amount of work that is beyond the scope of the present study. See the section and lines below for changes:

- Section 'Recombination of chromosomes between strains and species': pages 10-11, lines 261-280.

4. Figure 2B and C – under both growth conditions the three top performing strains are consistent. Do the authors have any hypotheses on why these combinations are improving the performance compared to the wild type BY and RM?

For a complex trait, the specific phenotype exhibited by a genotype will depend on the additive and epistatic effects of causative alleles. Presumably the specific combinations of alleles in these strains produce these phenotypes. The challenge is again that our mapping resolution does not permit deeper insight. We do not know for example how many specific causative polymorphisms are present across all segments or how they interact with each other and the rest of the genome.

5. Figure 2B and C – is the number of experimental replicates missing here (is it 12?)

Presumably this refers to 2C and D. 12 replicates is correct. We have updated the figure legend to make it more clear. Specifically, we added the parenthetical 'n = 12' within the legend.

6. Figure 3B – the strain with segments 1-3-2 has twice the length of doubling time. Do authors have any explanation for this particular orientation causing a growth defect?

Because all of the strains with chromosome restructurings have the same gene content, we hypothesized that one or more changes in gene expression are likely responsible for the 1-3-2 strain's growth defect, as well as for the less reduced growth of the 3-1-2 strain. We also hypothesized that the most likely causative genes would be ones near the junctions between segments, as changes in their chromosomal contexts might affect their expression. Evaluating all segment orders revealed junctions unique to these chromosome configurations involving the non-essential genes *MYO8*, *NUP60*, and *SNC1*, all of which are known to cause growth defects when overexpressed. Using qPCR, we analyzed all three genes in both of these strains with growth defects, as well as in the 1-2-3 control strain. This revealed that all three genes showed overexpression in the 1-3-2 strain and that one of the genes showed overexpression in the 3-1-2 strain. These results suggest that gene overexpression may be the cause for the 1-3-2 growth defect. See the new version of Figure 3 and the sections and lines below for changes:

- Section 'Chromosome restructuring': pages 12 and 13, lines 315 -331.

7. Methods, Cloning of natural genomic segments – can authors elaborate on how three gRNAs were designed to cut each site of the segment? What would be the distance between the 3 generated cuts? How does it affect the length of the region used for homologous recombination if some gRNAs are not functional?

As now discussed more in the Methods and the main text section 'A system for cloning and reprogramming natural DNA for assembly', we employed three gRNAs for each side of a target segment (six total per segment), choosing gRNAs with the highest predicted on-target scores within 1 kb of the homology sites used for cloning. We selected gRNAs based on the built-in gRNA design tool present on Benchling, requiring on-target scores equal or higher than 75 on a 0-100 scale. If genes were present within the 1 kb regions, preference was given to gRNAs that had lower on-target scores but were outside known coding or regulatory regions. The maximum distance between gRNAs targeting the same side of a segment was ~200 bp. We did not conduct experiments that could relate individual gRNA position and efficacy to homologous recombination. By picking three gRNAs roughly targeting the same location, we avoided issues associated with non-efficacious gRNAs. This choice was based on preliminary experiments that showed our cloning method did not work well without CRISPR cutting. See the sections and lines below for changes:

- Section 'Results and Discussion': page 5, lines 99 -110.
- Section 'In vitro transcription of gRNAs': pages 2 and 3, lines 63-69.

8. Methods, Amplification of cloned segments – “inspecting spheroplasts every 15 min” – how was this done? Please explain more?

50 ml of yeast cells growing overnight were washed with water and centrifuged. The cells were resuspended in Y1 buffer (1M Sorbitol, 100 mM EDTA pH 8.0, 14 mM beta mercaptoethanol) and 1,000 units of Lyticase. Cells were incubated at 30°C for at least 15 minutes. Spheroplast formation was inspected every 15 min under an optical microscope. Spheroplast cells are larger and more rounded than regular yeast cells. Reactions were stopped when the majority of analyzed cells showed the features of spheroplasts. Typically this took between 45 min and 1 hr. See the section and lines below for changes:

- Section 'Amplification of cloned segments' from Methods: page 4, line 105-113.

9. Methods, Chromosome assembly – “co-transforming >500 ng” – can you please describe how the amount of DNA of each segment was determined?

A few, early transformations were conducted with 1 ug, but we later found that 500 ng per segment was sufficient. Thus, nearly all work was done with 500 ng. We have eliminated the '>' from the Methods to avoid confusing language suggesting that we calibrated the amount of each segment we used based on criteria such as size.

10. Methods ONT library prep – authors claim multiplexing up to 10 strains for the whole genome sequencing on MinION. What was the minimal depth of the sequencing reads for each strain in this multiplex experiment?

We required a minimum average per site coverage of 10x for structure calling and 25x for SNP calling. In multiple cases, a strain had to be sequenced on a second subsequent flow cell because its initial coverage was insufficient. This is now made more clear in the Methods. See the section and lines below for changes:

- Section 'Sequence analysis of synthetic chromosomes' from Methods: page 11, line 336-341.

11. Supplementary Figure 8 – this figure will be more readable if ONT reads are not included and the figure only presents the combinations of segments.

We appreciate this comment. However, we refrained from modifying the figure because we believe it is important to show that these are sequence-validated chromosome assemblies.

12. Supplementary Table 1 and 19 – It would be more easy for readers if annotated sequences of plasmids were provided (e.g. Genbank files or Benchling links) rather than just giving a sequence in table.

We deposited the plasmids in Addgene. The links to the plasmids are in Supplementary Table 1. The Genbank files for the plasmids are available on Addgene via the links.

13. Why did the authors decide to transform in vitro transcribed gRNAs rather than express gRNAs in vivo? This is not a common choice for yeast research.

As discussed in the main text, we used IVT gRNAs instead of more conventional gRNA plasmids because the former allows PCR products to be used as templates, eliminating assembly and verification of gRNA plasmids. IVT gRNAs make it possible to use several distinct gRNAs for the same target with little added work, removing the need to screen individual gRNAs for efficacy. See the section and lines below for changes:

- Section 'Results and Discussion': page 5, lines 99 -110.

Reviewer #2 (Remarks to the Author):

Coradini et al. described a CRISPR-Cas9 assisted gap-repair cloning strategy that efficiently clones native yeast chromosome fragments ranging from 3.8 kb to 64 kb. Thanks to the unique flanking adaptors, the authors were able to assemble the cloned fragments into a larger construct in a defined order in the subsequent step. Taking advantage of this flexibility, the authors first assembled 27 possible Chr.I recombinants consisting of three fragments cloned from three *Saccharomyces* strain/species in their native order. Second, the authors reorganized the fragment orders and assembled 5 non-natural Chr.I structures using the 3 intermediate Chr.I fragments cloned from the *S.cerevisiae* BY strain. In both cases, the doubling time was measured as a readout. Lastly, the authors intended to delete 10 regions in the genome using their method, which required cloning 11 intermediate fragments, then seemed combined the assembly step with native Chr.I elimination step in one transformation, but failed to recover colonies with all 10 deletions. After examining a clone with 9 deletions, they found a potential important genetic interaction between SYN8 and other deleted genes. This manuscript provides a cost-effective way to “shuffle” a native yeast chromosome by design, but the applications of this technique seem somewhat limited. These are the major concerns to be addressed:

1. The authors cloned yeast chromosome I as a circular BAC/YAC plasmid, but they did not determine the plasmid copy number. A plasmid copy number quantification assay should be performed before analyzing the strain fitness. This is important because the CEN/ARS sequences seem to lead to higher than 1 copy/cell e.g. in the pRS vector series. Ideally, comparison can be made to the native copy number before the chromosome destabilization step.

Based on whole-genome sequencing, we did not find that synthetic ChrI molecules are present at more than a single copy per cell. For example, in the euploid containing *S. paradoxus* ChrI, ChrI exhibits comparable per site coverage to other chromosomes (see red dot in panel a of figure below, which is also now included as Supplementary Figure 5). This coverage analysis is complicated by the fact that we did high molecular weight DNA extractions, which tend to be biased against segments of shorter chromosomes (see panel b in figure below). *S. paradoxus* ChrI shows similar coverage to other small, native chromosomes that are present at a single copy in the cell (i.e., ChrIII and ChrVI).

Additionally, in response to Reviewer 2, we also checked the aneuploid containing the full *S. paradoxus* ChrI and found evidence that the synthetic ChrI showed slightly lower coverage than the native copy (86%). In the aneuploid, a variety of factors can potentially cause the synthetic chromosome to show some instability. For example, all essential genes also have a copy on the native chromosome and marker expression may produce enough protein product for a daughter cell to transiently survive even if it does not have a copy of the synthetic chromosome containing the markers. This is why we show the data from the euploid instead. See also the sections and lines below for changes:

- Section 'Initial assembly of a chromosome using CReATiNG': pages 8 and 9, lines 203 -205.
- Section 'Sequence analysis of synthetic chromosomes' from Methods: page 11, lines 336-341.

2. In Figure 2 and 3, where are the boundaries of the three fragments, how they are defined? Is it possible that reorganizing might simply have altered the gene expression near the fragment boundaries, RNA profiling can help sort this out?

As discussed in the main text section 'Initial assembly of a chromosome using CReATiNG', in silico, we divided ChrI into three non-overlapping segments between 51 and 64 kb, which contained the entire chromosome except the centromere, subtelomeres, and telomeres. The segments were each designed to contain roughly one-third of ChrI and to avoid disruption of annotated functional elements. We cloned segments 2 and 3 in a manner that excluded the natural ChrI centromere, as we supply a synthetic centromere cassette containing *CEN6*, as well as drug resistance markers for both yeast (*kanMX*) and *E. coli* (*ampR*), during chromosome assembly. We excluded subtelomeres from all work, as they are completely dispensable and highly variable across *Saccharomyces* strains and species. Telomeres were also excluded because they are not amenable to cloning and we initially assemble chromosomes as circular, rather than linear, molecules.

We agree that reorganizing these segments may affect expression of genes on their boundaries. Please see the response to Reviewer 1 Question 6 where this same point is addressed.

3. In Figure 4, the adaptor sequences are left behind in the genome, it seems the author could make the deletion scarless by using the next fragment's sequence as adaptor – why are the special adaptors even needed? Also, if the authors want to make the multi-gene deletion strategy generalizable, they should design a selection strategy that allows replacement of the native chromosome segment with the assembled segment. In this manuscript the authors relied on the deletion of *ADE1*, which is not an efficient way to screen for the desired construct. Wouldn't a simple drug resistance marker or the like be much more efficient?

In our experience, assembling *Saccharomyces* DNA inside *Saccharomyces* cells does not work well without adapters. We suspect this is because linear DNA fragments can recombine with homologous sequences in native chromosomes in vivo. By contrast, assembling non-*Saccharomyces* DNA inside yeast works well and does not require such adapters, presumably because homologous sequences are not present in the genome.

The selection strategy applied for the multi-gene deletion was the same strategy used for chimeras and restructured chromosomes. We first identified likely correct assemblies by selection for *HIS3* on the vector and *kanMX* in the centromere cassette. We then performed native chromosome elimination by native chromosome centromere destabilization and selection against *URA3*. The *ADE1* deletion, which causes colonies to turn red, was only used as an additional marker because it was present on the native ChrI but not in our synthetic ChrI. Effectively *ADE1* was a bonus marker we got with no added effort. See the sections and lines below for changes:

- Section 'Initial assembly of a chromosome using CReATiNG': page 8, lines 180-186.
- Section 'Multiplex gene deletion and streamlining' : page 14, lines 364-365.

4. The annotation of the *SYN8* gene on SGD claims that the null mutation has increased competitive fitness, which contradicts what this manuscript reports, the authors should provide some thoughts on why their conclusion is contradictory. Also, the authors could search previous genetic interaction studies to see if there are known important interactions between *SYN8* and other genes on Chr. 1.

SYN8 shows a different effect on growth when deleted at the same time as other chromosomal loci as opposed to when deleted individually. The explanation must be a genetic interaction with the multiplex deletion background, but the nature of this interaction is not clear. We looked at potential *SYN8* interactors among genes on ChrI using the *Saccharomyces* Genome Database (26, 40). There were zero genetic interactors, a single physical interactor at the mRNA level (*Ccr4*), and a single physical interactor at the protein level (*Snc1*). However, neither *CCR4* nor *SNC1* were eliminated in the multiplex deletion experiment. While it is conceivable that a change in their expression as a consequence of the multiplex deletion might produce a novel genetic interaction with *SYN8*, another possibility is that two or more genes that were simultaneously deleted interact with *SYN8* in a previously unidentified higher-order genetic interaction. Disentangling how *SYN8* interacts with the genetic background is an interesting, but complex challenge beyond the scope of the current manuscript. Most of this paragraph was also added to section 'Multiplex gene deletion and streamlining' page 15, lines 397-408. See also pages 14 and 15, lines 385-395 and page 15, lines 410-417 for additional changes.

5. Title : While CREATE is a cute name, this isn't creating a sequence from scratch; Better to use "recombinant chromosome" instead of "synthetic chromosome" to describe the strains they generated.

We understand how opinions on the naming of methods can vary and we appreciate the suggestion. However, we do believe that the current name of the method—Cloning, Reprogramming, and Assembling Tiled Natural Genomic DNA (CReATiNG)—is appropriate.

Reviewer #3 (Remarks to the Author):

In this manuscript, the authors reported a method for constructing synthetic chromosomes from natural components in yeast, called CReATiNG. They demonstrated the approach by native chromosome replacement, chromosome structure modification and multiplex gene deletion. The

advantage of the approach is its feasibility and convenient operation to obtain a long segment of natural DNA when compared with PCR-based or DNA synthesis-based methods.

However, my major concern is that similar studies have achieved cloning of natural segment, such as CATCH, reported by Wenjun Jiang et al. and CAPTURE, reported by Behnam Enghiad et al. The authors should describe the differences or improvements between CReATiNG and these two methods. In addition, the method is highly depended on the ability of homologous recombination of the host, it is hard to be applied in other organisms.

Thank you for pointing this out. We should have covered it in the original submission. As we now mention in the Conclusion, CReATiNG is highly efficient and shows low sequence constraints relative to in vitro approaches. For example, most in vitro approaches for cloning natural DNA rely on Gibson assembly between a target segment and a vector, but such reactions depend on the overlap between the segment and the vector being within a few nucleotides of the end of the segment. With CReATiNG, the overlaps between a target segment and the homology arms in a vector do not need to be immediately at the end of a segment. This provides flexibility in where CRISPR cut sites can be located, as they do not need to be directly adjacent to homology arms. Regarding the use of CReATiNG in other organisms, we believe that will depend on factors such as the innate ability of the organism to undergo homologous recombination. However, we also stress that CReATiNG can potentially be used for reengineering any synthetic chromosome or genome stored in the yeast nucleus. For example, as most synthetic bacterial genomic segments or complete genomes are synthesized by assembly in the yeast nucleus, CReATiNG could be used to reengineer these genomes. Please see changes on Section 'Conclusion' page 17 and 18, lines 440-457.

The other major concern I have is about the approach. Is that the recombination efficiency would decrease rapidly with increased sizes and number of segments. Are the three segments used for the chromosome assembly with the same selection marker? The segments were released from the plasmid by I-SceI in vitro. The following steps were the same with standard TAR-based assembly. Is it possible to finish these steps in vivo? Because when the size of the segment increases, it is more difficult to extract and purify the required segments from the yeast cell.

We agree that the number and size of segments being assembled could impact assembly efficiency and may complicate steps such as extraction and purification. We transform segments into yeast as linear DNA molecules that do not have markers. The markers are simply present on the vector and the centromere cassette. This seems to work well, even for >10 segments.

Regarding doing the final steps of CReATiNG in vivo, there are some limitations. First, markers will be limiting in this situation as each segment will require a distinct marker on its vector. Second, many segments are kept as large circular DNA molecules prior to I-SceI digestion. We have found that co-transforming a number of large circular DNA molecules into yeast works much less well than large linear DNA molecules. Third, while work has been done to show I-SceI can be used for linearizing individual molecules in vivo by single or double cutting, to our knowledge it has not been shown that it will effectively linearize many distinct molecules in the same cell in the same manner in vivo.

We have not had any issues to date working with segments up to the 150-200 kb range. Of course this requires standard precautions for dealing with high molecular weight DNA, such as using wide bore tips and not vortexing samples. We would argue that handling DNA much beyond the 150-200 kb range is usually unnecessary and involves a largely distinct set of procedures, such as agarose plugs or even fusing cells instead of extracting molecules. Molecules of this very large size range are beyond the scope of the current paper.

Other minor issues:

1. In Figs. 2e and 2f, the variations within the groups were much larger than those between them. I don't understand what the dots represent in each group. Was one dot represented one strain with a recombinant chromosome? It was rather imprudent to get the conclusion that "all three ChrI segments contribute to growth variation across temperatures". The conclusion should be more specific after additional validate experiments.

There were 27 different chromosomes examined in this part of the paper. Each dot represents the mean phenotype for a particular chromosome (e.g., BY-BY-BY, BY-Spar-RM, etc.), as determined by averaging 12 replicate cultures for that chromosome. Each allele of a segment occurs in nine different ChrI haplotypes. Thus, for a given segment, there are nine points, each representing the mean phenotype of a strain carrying the stated allele of the segment but otherwise having a different haplotype. In genetics, it is standard to calculate the phenotypic effect of an allele by averaging across genetic backgrounds carrying the allele, as we have done here. The variance among points reflects the variability in phenotypes among strains that carry the same allele but otherwise have different haplotypes. Such variability is expected because while these strains carry the same allele of a segment, they differ elsewhere on ChrI. See the section and lines below for changes:

- Section 'Recombination of chromosomes between strains and species': page 10 and 11, lines 261-280.

2. Whether the gene compositions and organizations of the three segments from three species were different as they showed different length (Fig. S7). If there were differences, it may be related to the phenotype variations.

As discussed in the response to Reviewer 1 Question 3, at the end of the section 'Recombination of chromosomes between strains and species', we now explicitly discuss this point. There are a few gene content differences between *S. cerevisiae* and *S. paradoxus* segment 3. While we cannot say with certainty that these gene content differences between the species are responsible for the effect of segment 3, we at least now point them out. We reiterate that the thermotolerance difference between *S. cerevisiae* and *S. paradoxus* is known to be very genetically complex, so determining the exact causative genes is likely to be a significant effort beyond the scope of the current paper. See the section and lines below for changes:

- Section 'Recombination of chromosomes between strains and species': pages 10-11, lines 261-280.

3. It was interesting that the 1-3-2 configurations showed growth reductions compared to the natural 1-2-3 configuration. More experiments such as gene expression profile in these strains could be provided to explain this growth deflection.

Please see the response to Reviewer 1 Question 6 where this same point is addressed.

4. I do not think the last application of CReATiNG was a good example as its performance was more complicate than PCR-based assembly. In addition, CReATiNG cannot provide a solution for highly multiplexed deletion as the designs for all the segments were knowledge-based. The success of deletion of SYN8 in the multiple deletion strain also proved that it was possible to construct the targeted strain with all sequence deletion. However, the wild-type copy of Chr I hindered the synthetic chromosomes generation.

As we now show in the Results and Discussion, we were in fact able to recover the complete assembly with the originally intended multiplex deletion design. While revising the paper, we hypothesized that

the reason we likely did not recover this intended design at first was due to its substantially reduced growth. To recover the intended multiplex deletion strain, we redid the assembly transformation and screened transformants from the chromosome assembly as aneuploids that still contained the native copy of ChrI. The assumption was that growth defects caused by multiplex deletion should be at least partly complemented in aneuploids. Consistent with this hypothesis, we were able to find an aneuploid with the intended assembly. After identifying an aneuploid with the correct assembly, we then performed native chromosome elimination. This successfully recovered a euploid strain with the intended multiplex deletion design. Please see the paragraphs added to section 'Multiplex gene deletion and streamlining' pages 14 and 15, lines 385-417 and also Supplementary Figure 15.

Regarding the potential to make this multiplex deletion design by PCR assembly, to our knowledge, a ~148 kb construct would be very difficult to produce by PCR assembly and, if it were possible, use of PCR would increase the occurrence of mutations (~1 every 10 kb). A molecule of this size can also be difficult to work with in vitro.

Lastly, we would emphasize that the ability to recover chromosomes that result from recombination between a synthetic chromosome and its natural counterpart is a benefit of CReATiNG, not a weakness. A major challenge in synthetic genomics is design flaws ('bugs') that may not necessarily be predictable in advance. As in the *SYN8* example, such bugs will likely be a major impediment in efforts to streamline and minimize the genomes of organisms with high levels of genetic redundancy, such as *S. cerevisiae*. However, CReATiNG can be used to identify such bugs, making it possible to pursue chromosome streamlining and minimization.

Reviewers' Comments:

Reviewer #1:

Remarks to the Author:

I am happy to say that the revised versions of the manuscript is improved and now much easier to follow in terms of a method, with appropriate description of the steps and the tools used. The lack of description of the method was my main concern in the initial review my postdoc and I did of this paper, and this is now addressed with the new version. All minor errors and clarifications have now been addressed too.

Congratulations to the authors on an interesting and well described study.

Reviewer #2:
Remarks to the Author:

Black text – R1 comments
Blue text – R1 responses
Red text – R2 comments

The authors addressed my major concerns, but some minor points still need to be addressed.

Reviewer #2 (Remarks to the Author):

Coradini et al. described a CRISPR-Cas9 assisted gap-repair cloning strategy that efficiently clones native yeast chromosome fragments ranging from 3.8 kb to 64 kb. Thanks to the unique flanking adaptors, the authors were able to assemble the cloned fragments into a larger construct in a defined order in the subsequent step. Taking advantage of this flexibility, the authors first assembled 27 possible Chr.I recombinants consisting of three fragments cloned from three *Saccharomyces* strain/species in their native order. Second, the authors reorganized the fragment orders and assembled 5 non-natural Chr.I structures using the 3 intermediate Chr.I fragments cloned from the *S.cerevisiae* BY strain. In both cases, the doubling time was measured as a readout. Lastly, the authors intended to delete 10 regions in the genome using their method, which required cloning 11 intermediate fragments, then seemed combined the assembly step with native Chr.I elimination step in one transformation, but failed to recover colonies with all 10 deletions. After examining a clone with 9 deletions, they found a potential important genetic interaction between *SYN8* and other deleted genes. This manuscript provides a cost-effective way to “shuffle” a native yeast chromosome by design, but the applications of this technique seem somewhat limited. These are the major concerns to be addressed:

1. The authors cloned yeast chromosome I as a circular BAC/YAC plasmid, but they did not determine the plasmid copy number. A plasmid copy number quantification assay should be performed before analyzing the strain fitness. This is important because the CEN/ARS sequences seem to lead to higher than 1 copy/cell e.g. in the pRS vector series. Ideally, comparison can be made to the native copy number before the chromosome destabilization step.

Based on whole-genome sequencing, we did not find that synthetic ChrI molecules are present at more than a single copy per cell. For example, in the euploid containing *S. paradoxus* ChrI, ChrI exhibits comparable per site coverage to other chromosomes (see red dot in panel a of figure below, which is also now included as Supplementary Figure 5). This coverage analysis is complicated by the fact that we did high molecular weight DNA extractions, which tend to be biased against segments of shorter chromosomes (see panel b in figure below). *S. paradoxus* ChrI shows similar coverage to other small, native chromosomes that are present at a single copy in the cell (i.e., ChrIII and ChrVI).

Additionally, in response to Reviewer 2, we also checked the aneuploid containing the full *S. paradoxus* ChrI and found evidence that the synthetic ChrI showed slightly lower coverage than the native copy (86%). In the aneuploid, a variety of factors can potentially cause the synthetic chromosome to show

some instability. For example, all essential genes also have a copy on the native chromosome and marker expression may produce enough protein product for a daughter cell to transiently survive even if it does not have a copy of the synthetic chromosome containing the markers. This is why we show the data from the euploid instead. See also the sections and lines below for changes:

- Section 'Initial assembly of a chromosome using CReATiNG': pages 8 and 9, lines 203 -205.
- Section 'Sequence analysis of synthetic chromosomes' from Methods: page 11, lines 336-341.

The authors reanalyzed the ONT sequencing data, and showed the “synthetic” ChrI had similar coverage relative to other chromosomes in the same strain, which is one way to address the concern. Although I think a more appropriate comparison would be “synthetic” ChrI coverage in syn strain vs. native ChrI coverage in parental strain (Figure 1d, the 3rd strain vs the 1st strain).

2. In Figure 2 and 3, where are the boundaries of the three fragments, how they are defined? Is it possible that reorganizing might simply have altered the gene expression near the fragment boundaries, RNA profiling can help sort this out?

As discussed in the main text section 'Initial assembly of a chromosome using CReATiNG', in silico, we divided ChrI into three non-overlapping segments between 51 and 64 kb, which contained the entire chromosome except the centromere, subtelomeres, and telomeres. The segments were each designed to contain roughly one-third of ChrI and to avoid disruption of annotated functional elements. We cloned segments 2 and 3 in a manner that excluded the natural ChrI centromere, as we supply a synthetic centromere cassette containing CEN6, as well as drug resistance markers for both yeast (kanMX) and E. coli (ampR), during chromosome assembly. We excluded subtelomeres from all work, as they are completely dispensable and highly variable across *Saccharomyces* strains and species. Telomeres were also excluded because they are not amenable to cloning and we initially assemble chromosomes as circular, rather than linear, molecules.

We agree that reorganizing these segments may affect expression of genes on their boundaries. Please see the response to Reviewer 1 Question 6 where this same point is addressed.

The authors addressed this comment by performing RT-qPCR of *MYO4* in 1-3-2 and 3-1-2 strains comparing to 1-2-3, as well as *SNC1* and *NUP60* in 1-3-2, which had 68% growth reduction comparing to 1-2-3. The overexpression of these 3 genes cause decreased growth rate according to previous studies.

3. In Figure 4, the adaptor sequences are left behind in the genome, it seems the author could make the deletion scarless by using the next fragment's sequence as adaptor – why are the special adaptors even needed? Also, if the authors want to make the multi-gene deletion strategy generalizable, they should design a selection strategy that allows replacement of the native chromosome segment with the assembled segment. In this manuscript the authors relied on the deletion of *ADE1*, which is not an efficient way to screen for the desired construct. Wouldn't a simple drug resistance marker or the like be much more efficient?

In our experience, assembling *Saccharomyces* DNA inside *Saccharomyces* cells does not work well without adaptors. We suspect this is because linear DNA fragments can recombine with homologous sequences in native chromosomes in vivo. By contrast, assembling non-*Saccharomyces* DNA inside yeast works well and does not require such adaptors, presumably because homologous sequences are not present in the genome.

The selection strategy applied for the multi-gene deletion was the same strategy used for chimeras and restructured chromosomes. We first identified likely correct assemblies by selection for *HIS3* on the vector and *kanMX* in the centromere cassette. We then performed native chromosome elimination by native chromosome centromere destabilization and selection against *URA3*. The *ADE1* deletion, which causes colonies to turn red, was only used as an additional marker because it was present on the native ChrI but not in our synthetic ChrI. Effectively *ADE1* was a bonus marker we got with no added effort. See the sections and lines below for changes:

- Section 'Initial assembly of a chromosome using CReATiNG': page 8, lines 180-186.

- Section 'Multiplex gene deletion and streamlining' : page 14, lines 364-365.

The authors addressed my concern (I thought the authors replaced one region of the chromosome, so that the assembled fragment need to be integrated to the chromosome, but the 11 fragments actually covered the whole ChrI, so they can eliminate the native ChrI by counterselecting *URA3*).

4. The annotation of the *SYN8* gene on SGD claims that the null mutation has increased competitive fitness, which is contradicts what this manuscript reports, the authors should provide some thoughts on why their conclusion is contradictory. Also, the authors could search previous genetic interaction studies to see if there are known important interactions between *SYN8* and other genes on Chr. 1.

SYN8 shows a different effect on growth when deleted at the same time as other chromosomal loci as opposed to when deleted individually. The explanation must be a genetic interaction with the multiplex deletion background, but the nature of this interaction is not clear. We looked at potential *SYN8* interactors among genes on ChrI using the Saccharomyces Genome Database (26, 40). There were zero genetic interactors, a single physical interactor at the mRNA level (*Ccr4*), and a single physical interactor at the protein level (*Snc1*). However, neither *CCR4* nor *SNC1* were eliminated in the multiplex deletion experiment. While it is conceivable that a change in their expression as a consequence of the multiplex deletion might produce a novel genetic interaction with *SYN8*, another possibility is that two or more genes that were simultaneously deleted interact with *SYN8* in a previously unidentified higher-order genetic interaction. Disentangling how *SYN8* interacts with the genetic background is an interesting, but complex challenge beyond the scope of the current manuscript. Most of this paragraph was also added to section 'Multiplex gene deletion and streamlining' page 15, lines 397-408. See also pages 14 and 15, lines 385-395 and page 15, lines 410-417 for additional changes.

The authors added possible explanations regarding the discrepancies between SGD and their results about *SYN8*. From the manuscript, the authors didn't provide the sequencing validation of the 11 pieces BAC/YAC, it will be interesting to know whether the *SYN8* gene was inserted into the BAC/YAC during assembly, or did the insertion happen during the ChrI elimination. The authors probably have the answer already, but need to make it clear.

5. Title : While CREATE is a cute name, this isn't creating a sequence from scratch; Better to use "recombinant chromosome" instead of "synthetic chromosome" to describe the strains they generated.

We understand how opinions on the naming of methods can vary and we appreciate the suggestion. However, we do believe that the current name of the method—Cloning, Reprogramming, and Assembling Tiled Natural Genomic DNA (CReATiNG)—is appropriate.

The authors argued the CReATiNG title is appropriate, but my question is actually about the naming of the strains they made, sounds like "recombinant chromosome" is more appropriate.

Reviewer #3:

Remarks to the Author:

All my concerns have been addressed. I recommend publication at current format

Final Revision Response to Reviewers

Black text – R1 comments

Blue text – R1 responses

Red text – R2 comments

Orange text - R2 responses

The authors addressed my major concerns, but some minor points still need to be addressed.

Reviewer #2 (Remarks to the Author):

Coradini et al. described a CRISPR-Cas9 assisted gap-repair cloning strategy that efficiently clones native yeast chromosome fragments ranging from 3.8 kb to 64 kb. Thanks to the unique flanking adaptors, the authors were able to assemble the cloned fragments into a larger construct in a defined order in the subsequent step. Taking advantage of this flexibility, the authors first assembled 27 possible Chr.I recombinants consisting of three fragments cloned from three *Saccharomyces* strain/species in their native order. Second, the authors reorganized the fragment orders and assembled 5 non-natural Chr.I structures using the 3 intermediate Chr.I fragments cloned from the *S.cerevisiae* BY strain. In both cases, the doubling time was measured as a readout. Lastly, the authors intended to delete 10 regions in the genome using their method, which required cloning 11 intermediate fragments, then seemed combined the assembly step with native Chr.I elimination step

in one transformation, but failed to recover colonies with all 10 deletions. After examining a clone with 9 deletions, they found a potential important genetic interaction between SYN8 and other deleted genes. This manuscript provides a cost-effective way to “shuffle” a native yeast chromosome by design, but the applications of this technique seem somewhat limited. These are the major concerns to be addressed:

1. The authors cloned yeast chromosome I as a circular BAC/YAC plasmid, but they did not determine the plasmid copy number. A plasmid copy number quantification assay should be performed before analyzing the strain fitness. This is important because the CEN/ARS sequences seem to lead to higher than 1 copy/cell e.g. in the pRS vector series. Ideally, comparison can be made to the native copy number before the chromosome destabilization step.

Based on whole-genome sequencing, we did not find that synthetic ChrI molecules are present at more than a single copy per cell. For example, in the euploid containing *S. paradoxus* ChrI, ChrI exhibits comparable per site coverage to other chromosomes (see red dot in panel a of figure below, which is also now included as Supplementary Figure 5). This coverage analysis is complicated by the fact that we did high molecular weight DNA extractions, which tend to be biased against segments of shorter chromosomes (see panel b in figure below). *S. paradoxus* ChrI shows similar coverage to other small, native chromosomes that are present at a single copy in the cell (i.e., ChrIII and ChrVI).

Additionally, in response to Reviewer 2, we also checked the aneuploid containing the full *S. paradoxus* ChrI and found evidence that the synthetic ChrI showed slightly lower coverage than the native copy (86%). In the aneuploid, a variety of factors can potentially cause the synthetic chromosome to show some instability. For example, all essential genes also have a copy on the native chromosome and marker expression may produce enough protein product for a daughter cell to transiently survive even if it does not have a copy of the synthetic chromosome containing the markers. This is why we show the data from the euploid instead. See also the sections and lines below for changes:

- Section 'Initial assembly of a chromosome using CReATING': pages 8 and 9, lines 203 -205.
- Section 'Sequence analysis of synthetic chromosomes' from Methods: page 11, lines 336-341.

The authors reanalyzed the ONT sequencing data, and showed the “synthetic” ChrI had similar coverage relative to other chromosomes in the same strain, which is one way to address the concern. Although I think a more appropriate comparison would be “synthetic” ChrI coverage in syn strain vs. native ChrI coverage in parental strain (Figure 1d, the 3rd strain vs the 1st strain).

We analyzed the sequencing data for the parental strain BY4742 and added plots based on this analysis to Supplementary Figure 5, as suggested. The key insight from all of these coverage analyses is that the synthetic *S.*

paradoxus ChrI shows similar coverage to the other native, small chromosomes in the cell and to the native ChrI in the parental strain. These findings support the synthetic chromosome being present in a single copy in euploid cells.

2. In Figure 2 and 3, where are the boundaries of the three fragments, how they are defined? Is it possible that reorganizing might simply have altered the gene expression near the fragment boundaries, RNA profiling can help sort this out?

As discussed in the main text section 'Initial assembly of a chromosome using CReATiNG', in silico, we divided ChrI into three non-overlapping segments between 51 and 64 kb, which contained the entire chromosome except the centromere, subtelomeres, and telomeres. The segments were each designed to contain roughly one-third of ChrI and to avoid disruption of annotated functional elements. We cloned segments 2 and 3 in a manner that excluded the natural ChrI centromere, as we supply a synthetic centromere cassette containing CEN6, as well as drug resistance markers for both yeast (*kanMX*) and *E. coli* (*ampR*), during chromosome assembly. We excluded subtelomeres from all work, as they are completely dispensable and highly variable across *Saccharomyces* strains and species. Telomeres were also excluded because they are not amenable to cloning and we initially assemble chromosomes as circular, rather than linear, molecules.

We agree that reorganizing these segments may affect expression of genes on their boundaries. Please see the response to Reviewer 1 Question 6 where this same point is addressed.

The authors addressed this comment by performing RT-qPCR of *MYO4* in 1-3-2 and 3-1-2 strains comparing to 1-2-3, as well as *SNC1* and *NUP60* in 1-3-2, which had 68% growth reduction comparing to 1-2-3. The overexpression of these 3 genes cause decreased growth rate according to previous studies.

Reviewer 2 was satisfied with how we addressed their original comment.

3. In Figure 4, the adaptor sequences are left behind in the genome, it seems the author could make the deletion scarless by using the next fragment's sequence as adaptor – why are the special adaptors even needed? Also, if the authors want to make the multi-gene deletion strategy generalizable, they should design a selection strategy that allows replacement of the native chromosome segment with the assembled segment. In this manuscript the authors relied on the deletion of *ADE1*, which is not an efficient way to screen for the desired construct. Wouldn't a simple drug resistance marker or the like be much more efficient?

In our experience, assembling *Saccharomyces* DNA inside *Saccharomyces* cells does not work well without adapters. We suspect this is because linear DNA fragments can recombine with homologous sequences in native chromosomes in vivo. By contrast, assembling non-*Saccharomyces* DNA inside yeast works well and does not require such adapters, presumably because homologous sequences are not present in the genome.

The selection strategy applied for the multi-gene deletion was the same strategy used for chimeras and restructured chromosomes. We first identified likely correct assemblies by selection for *HIS3* on the vector and *kanMX* in the centromere cassette. We then performed native chromosome elimination by native chromosome centromere destabilization and selection against *URA3*. The *ADE1* deletion, which causes colonies to turn red, was only used as an additional marker because it was present on the native ChrI but not in our synthetic ChrI. Effectively *ADE1* was a bonus marker we got with no added effort. See the sections and lines below for changes:

- Section 'Initial assembly of a chromosome using CReATiNG': page 8, lines 180-186.

- Section 'Multiplex gene deletion and streamlining' : page 14, lines 364-365.

The authors addressed my concern (I thought the authors replaced one region of the chromosome, so that the assembled fragment need to be integrated to the chromosome, but the 11 fragments actually covered the whole ChrI, so they can eliminate the native ChrI by counterselecting *URA3*).

Reviewer 2 was satisfied with how we addressed their original comment.

4. The annotation of the SYN8 gene on SGD claims that the null mutation has increased competitive fitness, which is contradictory to what this manuscript reports, the authors should provide some thoughts on why their conclusion is contradictory. Also, the authors could search previous genetic interaction studies to see if there are known important interactions between SYN8 and other genes on Chr. 1.

SYN8 shows a different effect on growth when deleted at the same time as other chromosomal loci as opposed to when deleted individually. The explanation must be a genetic interaction with the multiplex deletion background, but the nature of this interaction is not clear. We looked at potential SYN8 interactors among genes on Chr1 using the Saccharomyces Genome Database (26, 40). There were zero genetic interactors, a single physical interactor at the mRNA level (Ccr4), and a single physical interactor at the protein level (Snc1). However, neither CCR4 nor SNC1 were eliminated in the multiplex deletion experiment. While it is conceivable that a change in their expression as a consequence of the multiplex deletion might produce a novel genetic interaction with SYN8, another possibility is that two or more genes that were simultaneously deleted interact with SYN8 in a previously unidentified higher-order genetic interaction. Disentangling how SYN8 interacts with the genetic background is an interesting, but complex challenge beyond the scope of the current manuscript. Most of this paragraph was also added to section 'Multiplex gene deletion and streamlining' page 15, lines 397-408. See also pages 14 and 15, lines 385-395 and page 15, lines 410-417 for additional changes.

The authors added possible explanations regarding the discrepancies between SGD and their results about SYN8. From the manuscript, the authors didn't provide the sequencing validation of the 11 pieces BAC/YAC, it will be interesting to know whether the SYN8 gene was inserted into the BAC/YAC during assembly, or did the insertion happen during the Chr1 elimination. The authors probably have the answer already, but need to make it clear.

As Reviewer 2 states, recombination between the synthetic chromosome and the native chromosome could potentially happen either during the assembly of a synthetic chromosome or during the transient aneuploid stage preceding native chromosome elimination. The only data we have that provides insight into which of these possibilities is correct is our work screening aneuploid colonies prior to native chromosome elimination. This was done to find a synthetic chromosome with all intended deletions, including SYN8. In this experiment, we found many aneuploid colonies with synthetic chromosomes that had recombined with the native chromosome, implying that substantial recombination happens during the chromosome assembly process itself. However, we cannot rule out the possibility that additional recombination can happen during native chromosome elimination. We have added the sentence 'These results also show that segments of the synthetic chromosome can recombine with the native chromosome during assembly and prior to native chromosome elimination.' to the relevant part of the Results.

5. Title : While CREATE is a cute name, this isn't creating a sequence from scratch; Better to use "recombinant chromosome" instead of "synthetic chromosome" to describe the strains they generated.

We understand how opinions on the naming of methods can vary and we appreciate the suggestion. However, we do believe that the current name of the method—Cloning, Reprogramming, and Assembling Tiled Natural Genomic DNA (CReATiNG)—is appropriate.

The authors argued the CReATiNG title is appropriate, but my question is actually about the naming of the strains they made, sounds like "recombinant chromosome" is more appropriate.

We appreciate Reviewer 2's point and now better understand it, but we regard synthetic chromosomes as a more appropriate descriptor than recombinant chromosomes. One reason Reviewer 2 may have their view is that we use homologous recombination in mitotic yeast cells to assemble chromosomes. However, it is standard in the synthetic genomics community to synthesize chromosomes for non-yeast species in this manner and typically the output chromosomes are described as synthetic chromosomes, not recombinant chromosomes. Another reason might be that we demonstrate a specific use case where we employ CReATiNG to recombine chromosomes between strains and species without mating or meiosis. Cells containing chromosomes generated in this manner are similar to recombinants produced by meiosis. Yet, this is just a single application of CReATiNG and some of the other

applications bear little resemblance to meiotic recombination. An additional reason Reviewer 2 might have their view is that there is a direct connection between recombinant DNA methods and synthetic genomic methods, and the size at which one refers to a molecule as recombinant DNA versus synthetic chromosome is not well-defined. While one could reasonably claim that all synthetic chromosomes are recombinant DNA molecules, it is standard in the synthetic genomics community to refer to whole chromosomes that have been assembled as synthetic chromosomes, not recombinant chromosomes. Lastly, we did not synthesize every nucleotide in the chromosomes we built. Reviewer 2 may have the view that this is why we should call output chromosomes recombinant and not synthetic. This is a valid point that we did consider both while drafting the manuscript and also while performing this final revision. However, our view is that chromosomes generated by CReATiNG do constitute synthetic chromosomes and that using the terminology synthetic chromosome is a more clear, accurate, and general descriptor. Our long response here reflects our gratitude for Reviewer 2's point and our desire to provide a thorough rationale for why we utilize the term synthetic chromosome.